# CROSS$Q$: BATCH NORMALIZATION IN DEEP REINFORCEMENT LEARNING FOR GREATER SAMPLE EFFICIENCY AND SIMPLICITY

**Aditya Bhatt** [*1,4]     **Daniel Palenicek** [*1,2]     **Boris Belousov** [1,4]     **Max Argus** [3]
**Artemij Amiranashvili** [3]        **Thomas Brox** [3]        **Jan Peters** [1,2,4,5]

[*]Equal contribution  [1]Intelligent Autonomous Systems, TU Darmstadt  [2]Hessian.AI  [3]University of Freiburg
[4]German Research Center for AI (DFKI)   [5]Centre for Cognitive Science, TU Darmstadt
`aditya.bhatt@dfki.de`, `daniel.palenicek@tu-darmstadt.de`

## ABSTRACT

Sample efficiency is a crucial problem in deep reinforcement learning. Recent algorithms, such as REDQ and DroQ, found a way to improve the sample efficiency by increasing the update-to-data (UTD) ratio to 20 gradient update steps on the critic per environment sample. However, this comes at the expense of a greatly increased computational cost. To reduce this computational burden, we introduce CrossQ: A lightweight algorithm for continuous control tasks that makes careful use of Batch Normalization and removes target networks to surpass the current state-of-the-art in sample efficiency while maintaining a low UTD ratio of 1. Notably, CrossQ does not rely on advanced bias-reduction schemes used in current methods. CrossQ's contributions are threefold: (1) it matches or surpasses current state-of-the-art methods in terms of sample efficiency, (2) it substantially reduces the computational cost compared to REDQ and DroQ, (3) it is easy to implement, requiring just a few lines of code on top of SAC.

## 1 INTRODUCTION

Sample efficiency is a crucial concern when applying Deep Reinforcement Learning (Deep RL) methods on real physical systems. One of the first successful applications of Deep RL to a challenging problem of quadruped locomotion was achieved using Soft Actor-Critic (SAC, Haarnoja et al. (2018a)), allowing a robot dog to learn to walk within 2h of experience (Haarnoja et al., 2018b). Subsequently, it was noted that the critic in SAC may be underfitted, as only a single gradient update step on the network parameters is performed for each environment step. Therefore, Randomized Ensembled Double Q-Learning (REDQ, Chen et al. (2021)) was proposed, which increased this number of gradient steps, termed update-to-data (UTD) ratio. In addition, Dropout Q functions (DroQ, Hiraoka et al. (2021)) improved the computational efficiency of REDQ while maintaining the same sample efficiency by replacing its ensemble of critics with dropout. This enabled learning quadruped locomotion in a mere 20min (Smith et al., 2022). Thus, REDQ and DroQ represent the state-of-the-art in terms of sample efficiency in Deep RL for continuous control.

Importantly, both REDQ and DroQ showed that naively increasing the UTD ratio of SAC does not perform well due to the critic networks' Q value estimation bias. Therefore, ensembling techniques were introduced for bias reduction (explicit ensemble in REDQ and implicit

Sample Efficiency

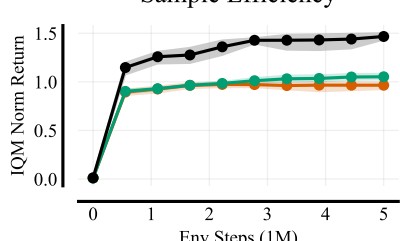

Computational Efficiency

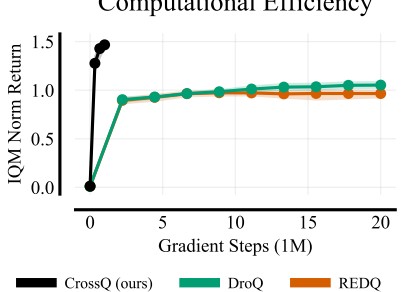

Figure 1: **CrossQ training performance aggregated over environments.** CrossQ is more sample efficient (top) while being significantly more computationally efficient (bottom) in terms of the gradient steps, thanks to a low UTD = 1. Following Agarwal et al. (2021), we normalize performance by the maximum of REDQ in each environment.

ensemble via dropout in DroQ), which allowed increasing the UTD to 20 critic updates per environment step. Higher UTD ratios improve sample efficiency by paying the price of increased computational cost, which manifests in higher wallclock time and energy consumption. It is, therefore, desirable to seek alternative methods that achieve the same or better sample efficiency at a lower computational cost, e.g., by using lower UTDs.

It turns out that even UTD = 1 can perform surprisingly well if other algorithmic components are adjusted appropriately. In this paper, we introduce **CrossQ**, a lightweight algorithm that achieves superior performance by removing much of the algorithmic design complexity that was added over the years, culminating in the current state-of-the-art methods. First, it *removes target networks*, an ingredient widely believed to slow down training in exchange for stability (Mnih et al., 2015; Lillicrap et al., 2016; Kim et al., 2019; Fan et al., 2020). Second, we find that *Batch Normalization* variants (Ioffe & Szegedy (2015); Ioffe (2017)), when applied in a particular manner, effectively stabilize training and significantly improve sample efficiency. This contradicts others' observations that it hurts the learning performance in Deep RL, e.g. Hiraoka et al. (2021). Third, Cross$Q$ uses *wider critic layers*, motivated by prior research on the ease of optimization of wider networks (Ota et al., 2021). In addition to the first two improvements, wider networks enable even higher returns.

**Contributions.** (1) We present the Cross$Q$ algorithm, which matches or surpasses the current state-of-the-art for model-free off-policy RL for continuous control environments with state observations in sample efficiency while being multiple times more computationally efficient; (2) By removing target networks, we are able to successfully accelerate off-policy Deep RL with Batch-Norm; (3) We provide empirical investigations and hypotheses for Cross$Q$'s success. Cross$Q$'s changes mainly pertain to the deep network architecture of SAC; therefore, our study is chiefly empirical: through a series of ablations, we isolate and study the contributions of each part. We find that Cross$Q$ matches or surpasses the state-of-the-art algorithms in sample efficiency while being up to $4\times$ faster in terms of wallclock time without requiring critic ensembles, target networks, or high UTD ratios. We provide the Cross$Q$ source code at github.com/adityab/CrossQ.

## 2 BACKGROUND

### 2.1 OFF-POLICY REINFORCEMENT LEARNING AND SOFT ACTOR-CRITIC

We consider a discrete-time Markov Decision Process (MDP, Puterman (2014)), defined by the tuple $\langle \mathcal{S}, \mathcal{A}, \mathcal{P}, \mathcal{R}, \rho, \gamma \rangle$ with state space $\mathcal{S}$, action space $\mathcal{A}$, transition probability $s_{t+1} \sim \mathcal{P}(\cdot | s_t, a_t)$, reward function $r_t = \mathcal{R}(s_t, a_t)$, initial state distribution $s_0 \sim \rho$ and discount factor $\gamma \in [0, 1)$. RL describes the problem of an agent learning an optimal policy $\pi$ for a given MDP. At each time step $t$, the agent receives a state $s_t$ and interacts with the environment according to its policy $\pi$. We focus on the Maximum Entropy RL setting (Ziebart et al., 2008), where the agent's objective is to find the optimal policy $\pi^*$, which maximizes the expected cumulative reward while keeping the entropy $\mathcal{H}$ high; $\arg\max_{\pi^*} \mathbb{E}_{s_0 \sim \rho} \left[ \sum_{t=0}^{\infty} \gamma^t (r_t - \alpha \mathcal{H}(\pi(\cdot | s_t))) \right]$. The action-value function is defined by $Q(s, a) = \mathbb{E}_{\pi, \mathcal{P}} \left[ \sum_{t=0}^{\infty} \gamma^t (r_t - \alpha \log \pi(a_t | s_t)) | s_0 = s, a_0 = a \right]$ and describes the expected reward when taking action $a$ in state $s$. Soft Actor-Critic (SAC, (Haarnoja et al., 2018a)) is a popular algorithm that solves the MaxEnt RL problem. SAC parametrizes the Q function and policy as neural networks and trains two independent versions of the Q function, using the minimum of their estimates to compute the regression targets for Temporal Difference (TD) learning. This *clipped double-Q* trick, originally proposed by Fujimoto et al. (2018) in TD3, helps in reducing the potentially destabilizing overestimation bias inherent in approximate Q-learning (Hasselt, 2010).

### 2.2 HIGH UPDATE-TO-DATA RATIOS, REDQ, AND DROQ

Despite its popularity among practitioners and as a foundation for other more complex algorithms, SAC leaves much room for improvement in terms of sample efficiency. Notably, SAC performs exactly one gradient-based optimization step per environment interaction. SAC's UTD = 1 setting is analogous to simply training for fewer epochs in supervised learning. Therefore, in recent years, gains in sample efficiency within RL have been achieved through increasing the UTD ratio (Janner et al., 2019; Chen et al., 2021; Hiraoka et al., 2021; Nikishin et al., 2022). Different algorithms, however, substantially vary in their approaches to achieving high UTD ratios. Janner et al. (2019)

```
1  def critic_loss(Q_params, policy_params, obs, acts, rews, next_obs):
2      next_acts, next_logpi = policy.apply(policy_params, next_obs)
3
4      # Concatenated forward pass
5      all_q, new_Q_params = Q.apply(Q_params,
6          jnp.concatenate([obs, next_obs]),
7          jnp.concatenate([acts, next_acts])
8      )
9      # Split all_q predictions and stop gradient on next_q
10     q, next_q = jnp.split(all_q, 2)
11     next_q = jnp.min(next_q, axis=0)    # min over double Q function
12     next_q = jax.lax.stop_gradient(next_q - alpha * next_logpi)
13     return jnp.mean((q - (rews + gamma * next_q))**2), new_Q_params
```

Figure 2: **CrossQ critic loss in JAX.** The CrossQ critic loss is easy to implement on top of an existing SAC implementation. One just adds the batch normalization layers into the critic network and removes the target network. As we are now left with only the critic network, one can simply concatenate observations and next observations, as well as actions and next actions along the batch dimension, perform a joint forward pass, and split up the batches afterward. Combining two forward passes into one grants a small speed-up thanks to requiring only one CUDA call instead of two.

uses a model to generate synthetic data, which allows for more overall gradient steps. Nikishin et al. (2022) adopt a simpler approach: they increase the number of gradient steps while periodically resetting the policy and critic networks to fight premature convergence to local minima. We now briefly outline the two high-UTD methods to which we compare CrossQ.

**REDQ.** Chen et al. (2021) find that merely raising SAC's UTD ratio hurts performance. They attribute this to the accumulation of the learned Q functions' estimation bias over multiple update steps—despite the clipped double-Q trick—which destabilizes learning. To remedy this bias more strongly, they increase the number of Q networks from two to an ensemble of 10. Their method, called REDQ, permits stable training at high UTD ratios up to 20.

**DroQ.** Hiraoka et al. (2021) note that REDQ's ensemble size, along with its high UTD ratio, makes training computationally expensive. They instead propose using a smaller ensemble of Q functions equipped with Dropout (Srivastava et al., 2014), along with Layer Normalization (Ba et al., 2016) to stabilize training in response to the noise introduced by Dropout. Called DroQ, their method is computationally cheaper than REDQ, yet still expensive due to its UTD ratio of 20.

## 3 THE CROSSQ ALGORITHM

In this paper, we challenge this current trend of high UTD ratios and demonstrate that we can achieve competitive sample efficiency at a much lower computational cost with a UTD = 1 method. CrossQ is our new state-of-the-art off-policy actor-critic algorithm. Based on SAC, it uses purely network-architectural engineering insights from deep learning to accelerate training. As a result, it ~~crosses out~~ much of the algorithmic design complexity that was added over the years and which led to the current state-of-the-art methods. In doing so, we present a much simpler yet more efficient algorithm. In the following paragraphs, we introduce the three design choices that constitute CrossQ.

### 3.1 DESIGN CHOICE 1: REMOVING TARGET NETWORKS

Mnih et al. (2015) originally introduced target networks to stabilize the training of value-based off-policy RL methods, and today, most algorithms require them (Lillicrap et al., 2016; Fujimoto et al., 2018; Haarnoja et al., 2018a). SAC updates the critics' target networks with Polyak Averaging

$$\boldsymbol{\theta}^\circ \leftarrow (1 - \tau)\boldsymbol{\theta}^\circ + \tau\boldsymbol{\theta}, \tag{1}$$

where $\boldsymbol{\theta}^\circ$ are the target network parameters, and $\boldsymbol{\theta}$ are those of the trained critic. Here $\tau$ is the *target network smoothing coefficient*; with a high $\tau = 1$ (equivalent to cutting out the target network), SAC training can diverge, leading to explosive growth in $\boldsymbol{\theta}$ and the $Q$ predictions. Target networks stabilize training by explicitly delaying value function updates, arguably slowing down online learning (Plappert et al., 2018; Kim et al., 2019; Morales, 2020).

SAC:          CrossQ (Ours):

$$Q_{\boldsymbol{\theta}}(\boldsymbol{S}_t, \boldsymbol{A}_t) = \boldsymbol{q}_t$$
$$Q_{\boldsymbol{\theta}^{\circ}}(\boldsymbol{S}_{t+1}, \boldsymbol{A}_{t+1}) = \boldsymbol{q}_{t+1}^{\circ}$$

$$Q_{\boldsymbol{\theta}}\left(\begin{bmatrix}\boldsymbol{S}_t \\ \boldsymbol{S}_{t+1}\end{bmatrix}, \begin{bmatrix}\boldsymbol{A}_t \\ \boldsymbol{A}_{t+1}\end{bmatrix}\right) = \begin{bmatrix}\boldsymbol{q}_t \\ \boldsymbol{q}_{t+1}\end{bmatrix}$$

$$\mathcal{L}_{\boldsymbol{\theta}} = (\boldsymbol{q}_t - \boldsymbol{r}_t - \gamma\,\boldsymbol{q}_{t+1}^{\circ})^2$$

$$\mathcal{L}_{\boldsymbol{\theta}} = (\boldsymbol{q}_t - \boldsymbol{r}_t - \gamma\,|\boldsymbol{q}_{t+1}|_{\mathtt{sg}})^2$$

Figure 3: SAC without BatchNorm in the critic $Q_{\boldsymbol{\theta}}$ (left) requires target $Q$ values $\boldsymbol{q}_{t+1}^{\circ}$ to stabilize learning. CrossQ with BatchNorm in the critic $Q_{\boldsymbol{\theta}}$ (right) removes the need for target networks and allows for a joint forward pass of both current and future values. Batches are sampled from the replay buffer $\mathcal{B}$: $\boldsymbol{S}_t, \boldsymbol{A}_t, \boldsymbol{r}_t, \boldsymbol{S}_{t+1} \sim \mathcal{B}$ and $\boldsymbol{A}_{t+1} \sim \pi_{\phi}(\boldsymbol{S}_{t+1})$ from the current policy. $|\cdot|_{\mathtt{sg}}$ denotes the `stop-gradient` operation.

Recently, Yang et al. (2021) found that critics with Random Fourier Features can be trained without target networks, suggesting that the choice of layer activations affects the stability of training. Our experiments in Section 4.4 uncover an even simpler possibility: using bounded activation functions or feature normalizers is sufficient to prevent critic divergence in the absence of target networks, whereas the common choice of `relu` without normalization diverges. While others have used normalizers in Deep RL before, we are the first to identify that they make target networks redundant. Our next design choice exploits this insight to obtain an even greater boost.

## 3.2    DESIGN CHOICE 2: USING BATCH NORMALIZATION

BatchNorm has not yet seen wide adoption in value-based off-policy RL methods, despite its success and widespread use in supervised learning (He et al., 2016; Santurkar et al., 2018), attempts at doing so have fared poorly. Lillicrap et al. (2016) use BatchNorm layers on the state-only representation layers in the DDPG critic but find that it does not help significantly. Others use BatchNorm in decoupled feature extractors for Deep RL networks (Ota et al., 2020; 2021), but not in critic networks. Hiraoka et al. (2021) report that using BatchNorm in critics causes training to fail in DroQ.

**We find using BatchNorm *carefully*, when *additionally* removing target networks, performs surprisingly well, trains stably, and is, in fact, algorithmically simpler than current methods.**

First, we explain why BatchNorm needs to be used *carefully*. Within the critic loss $[Q_{\boldsymbol{\theta}}(\boldsymbol{S}, \boldsymbol{A}) - (\boldsymbol{r} + \gamma Q_{\boldsymbol{\theta}^{\circ}}(\boldsymbol{S}', \boldsymbol{A}'))]^2$, predictions are made for two differently distributed batches of state-action pairs; $(\boldsymbol{S}, \boldsymbol{A})$ and $(\boldsymbol{S}', \boldsymbol{A}')$, where $\boldsymbol{A}' \sim \pi_{\phi}(\boldsymbol{S}')$ is sampled from the *current policy*, while $\boldsymbol{A}$ originates from old behavior policies.

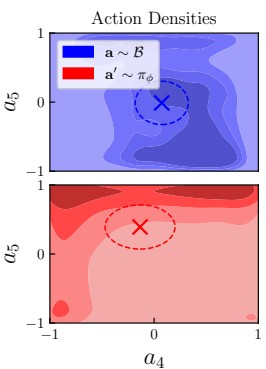

Figure 4: **Replay buffer and current policy actions are distributed differently.** Darker colors denote higher density. Estimated from a batch of $10^4$ transitions $(\boldsymbol{a}, \boldsymbol{s}') \sim \mathcal{B}$; $\boldsymbol{a}' \sim \pi_{\phi}(\boldsymbol{s}')$, after $3 \times 10^5$ training steps on `Walker2d`; $a_4$ and $a_5$ are random action dimensions.

Just like the target network, the BatchNorm parameters are updated by Polyak Averaging from the live network (Equation 1). The BatchNorm running statistics of the live network, which were estimated from batches of $(\boldsymbol{s}, \boldsymbol{a})$ pairs, will clearly not have *seen* samples $(\boldsymbol{s}', \pi_{\phi}(\boldsymbol{s}'))$ and will further not match their statistics. In other words, the state-action inputs evaluated by the target network will be out-of-distribution, given its mismatched BatchNorm running statistics. It is well known that the prediction quality of BatchNorm-equipped networks degrades in the face of such test-time distribution shifts (Pham et al., 2022; Lim et al., 2023).

Removing the target network provides an *elegant* solution. With the target network removed, we can concatenate both batches and feed them through the $Q$ network in a single forward pass, as illustrated in Figure 3 and shown in code in Figure 2. This simple trick ensures that BatchNorm's normalization moments arise from the union of both batches, corresponding to a 50/50 mixture of their respective distributions. Such normalization layers *do not* perceive the $(\boldsymbol{s}', \pi_{\phi}(\boldsymbol{s}'))$ batch as being out-of-distribution. This small change to SAC allows the safe use of BatchNorm and greatly accelerates training. We are not the only ones to identify this way of using BatchNorm to tackle the distribution mismatch; other works in supervised learning, e.g., Test-Time Adaptation (Lim et al.,

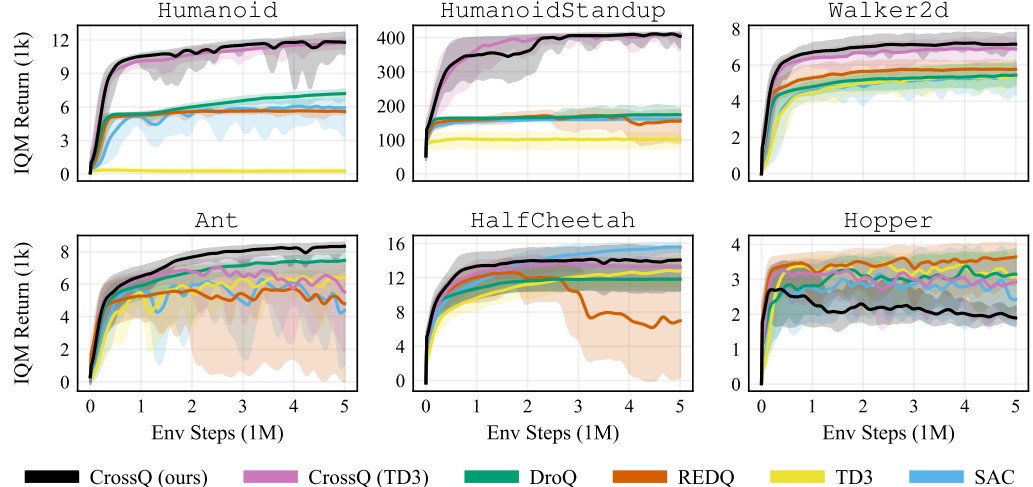

Figure 5: **Cross$Q$ sample efficiency.** Compared to REDQ and DroQ (UTD = 20) Cross$Q$ (UTD = 1) performs either comparably, better, or—for the more challenging `Humanoid` tasks—substantially better. These results directly transfer to TD3 as the base algorithm in Cross$Q$ (TD3). We plot *interquartile mean* (IQM) and 70% quantile interval of the episodic returns over 10 seeds.

2023), EvalNorm (Singh & Shrivastava, 2019), and *Four Things Everyone Should Know to Improve Batch Normalization* (Summers & Dinneen, 2020) also use mixed moments to bridge this gap.

In practice, Cross$Q$'s actor and critic networks use Batch Renormalization (BRN, Ioffe (2017)), an improved version of the original BN (Ioffe & Szegedy, 2015) that is robust to long-term training instabilities originating from minibatch noise. BRN performs batch normalization using the less noisy *running statistics* after a warm-up period, instead of noisy minibatch estimates as in BN. In the rest of this paper, all discussions with "BatchNorm" apply equally to both versions unless explicitly disambiguated by BN or BRN.

### 3.3 DESIGN CHOICE 3: WIDER CRITIC NETWORKS

Following Ota et al. (2021), we find that wider critic network layers in Cross$Q$ lead to even faster learning. As we show in our ablations in Section 4.4, most performance gains originate from the first two design choices; however, wider critic networks further boost the performance, helping to match or outperform REDQ and DroQ sample efficiency.

We want to stress again that **Cross$Q$**, a UTD = 1 method, ***does not use bias-reducing ensembles, high UTD ratios or target networks***. Despite this, it achieves its competitive sample efficiency at a fraction of the compute cost of REDQ and DroQ (see Figures 5 and 6). Note that our proposed changes can just as well be combined with other off-policy TD-learning methods, such as TD3, as shown in our experiments in Section 4.1.

## 4 EXPERIMENTS AND ANALYSIS

We conduct experiments to provide empirical evidence for Cross$Q$'s performance, and investigate:

1. Sample efficiency of Cross$Q$ compared to REDQ and DroQ;
2. Computational efficiency in terms of wallclock time and performed gradient step;
3. Effects of the proposed design choices on the performance via Q function bias evaluations;

And conduct further ablation studies for the above design choices. We evaluate across a wide range of continuous-control `MuJoCo` (Todorov et al., 2012) environments, with 10 random seeds each. Following Janner et al. (2019); Chen et al. (2021) and Hiraoka et al. (2021), we evaluate on the same four `Hopper`, `Walker2d`, `Ant`, and `Humanoid` tasks, as well as two additional tasks: `HalfCheetah` and the more challenging `HumanoidStandup` from Gymnasium (Towers et al., 2023). We adapted the JAX version of stable-baselines (Raffin et al., 2021) for our experiments.

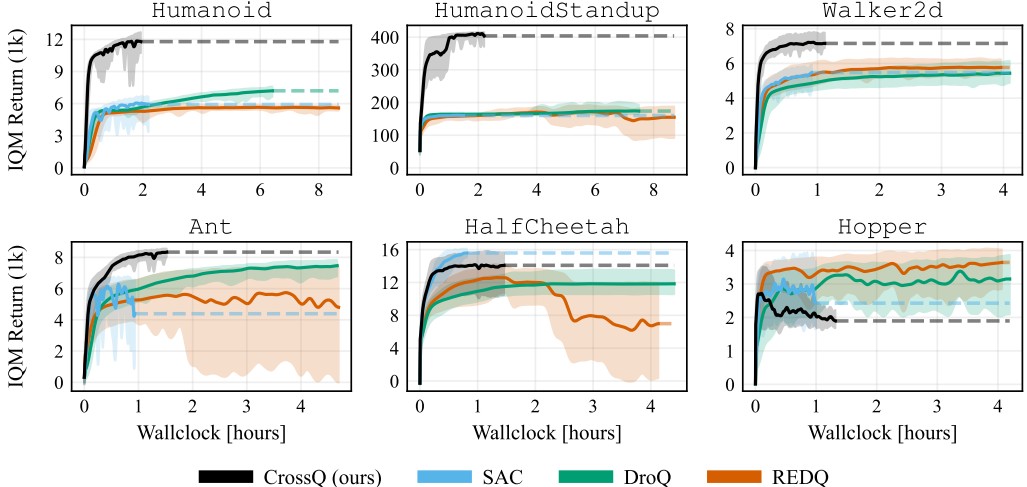

Figure 6: **Computational efficiency.** CrossQ trains an order of magnitude faster, taking only $5\%$ of the gradient steps, substantially saving on wallclock time. The dashed horizontal lines are visual aids to better compare the final performance after training for $5 \times 10^6$ environment steps. We plot IQM and $70\%$ quantile interval over 10 seeds. Appendix A.3 provides a table of wallclock times.

## 4.1 SAMPLE EFFICIENCY OF CROSSQ

Figure 5 compares our proposed CrossQ algorithm with REDQ, DroQ, SAC and TD3 in terms of their sample efficiency, i.e., average episode return at a given number of environment interactions. As a proof of concept, we also present CrossQ (TD3), a version of CrossQ which uses TD3 instead of SAC as the base algorithm. We perform periodic evaluations during training to obtain the episodic reward. From these, we report the mean and standard deviations over 10 random seeds. All subsequent experiments in this paper follow the same protocol.

This experiment shows that CrossQ matches or outperforms the best baseline in all the presented environments except on Ant, where REDQ performs better in the early training stage, but CrossQ eventually matches it. On Hopper, Walker, and HalfCheetah, the learning curves of CrossQ and REDQ overlap, and there is no significant difference. On the harder Humanoid and HumanoidStandup tasks, CrossQ and CrossQ (TD3) both substantially surpass all baselines.

## 4.2 COMPUTATIONAL EFFICIENCY OF CROSSQ

Figure 6 compares the computational efficiency of CrossQ to the baselines. This metric is where CrossQ makes the biggest leap forward. CrossQ requires $20\times$ fewer gradient steps than REDQ and DroQ, which results in roughly $4\times$ faster wallclock speeds (Table 2). Especially on the more challenging Humanoid and HumanoidStandup tasks the speedup is the most pronounced. In our view, this is a noteworthy feature. On the one hand, it opens the possibility of training agents in a truly online and data-efficient manner, such as in real-time robot learning. On the other hand, with large computing budgets CrossQ can allow the training of even larger models for longer than what is currently feasible, because of its computational efficiency stemming from its low UTD $= 1$.

## 4.3 EVALUATING $Q$ FUNCTION ESTIMATION BIAS

All methods we consider in this paper are based on SAC and, thus, include the clipped double-Q trick to reduce Q function overestimation bias (Fujimoto et al., 2018). Chen et al. (2021) and Hiraoka et al. (2021) stress the importance of keeping this bias even lower to achieve their high performances and intentionally design REDQ and DroQ to additionally reduce bias with explicit and implicit ensembling. In contrast, CrossQ outperforms both baselines without any ensembling. Could CrossQ's high performance be attributed to implicitly reducing the bias as a side effect of our design choices? Using the same evaluation protocol as Chen et al. (2021), we compare the

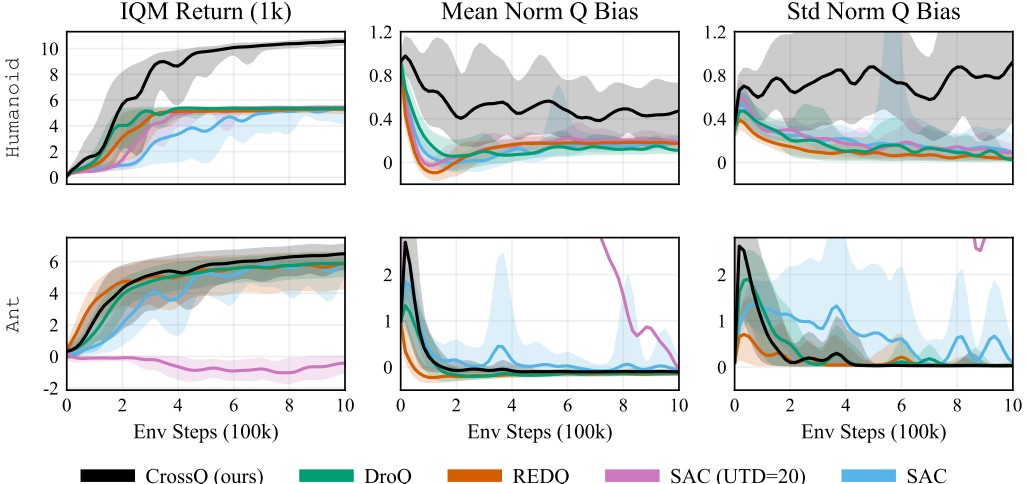

Figure 7: **Q estimation bias does not reliably influence learning performance**. Following the analysis of Chen et al. (2021), we plot the IQM and 70% quantile interval of the normalized Q function bias. REDQ generally has the least bias over 10 seeds. CrossQ matches or outperforms DroQ, REDQ and SAC while showing more Q function bias in all environments. The full set of environments is shown in Fig. 17 in the Appendix.

normalized Q prediction biases in Figure 4.3. Due to space constraints, here we show `Hopper` and `Ant` and place the rest of the environments in Figure 17 in the Appendix.

We find that REDQ and DroQ indeed have lower bias than SAC and significantly lower bias than SAC with UTD = 20. The results for CrossQ are mixed: while its bias trend exhibits a lower mean and variance than SAC, in some environments, its bias is higher than DroQ, and in others, it is lower or comparable. REDQ achieves comparable or worse returns than CrossQ while maintaining the least bias. As CrossQ performs better *despite* having—perhaps paradoxically—generally higher Q estimation bias, we conclude that the relationship between performance and estimation bias is complex, and one does not seem to have clear implications on the other.

## 4.4 ABLATIONS

We conduct ablation studies to better understand the impact of different design choices in CrossQ.

### 4.4.1 DISENTANGLING THE EFFECTS OF TARGET NETWORKS AND BATCHNORM

CrossQ changes SAC in three ways; of these, two explicitly aim to accelerate optimization: the removal of target networks, and the introduction of BatchNorm. Unfortunately, SAC without target networks diverges; therefore, to study the contribution of the first change, we need a way to compare SAC—divergence-free—*with and without target networks*. Fortunately, we find that such a way exists: according to our supplementary experiments in Appendix A.6, simply using bounded activation functions in the critic appears to prevent divergence. This is a purely empirical observation and an in-depth study regarding the influence of activations and normalizers on the stability of Deep RL is beyond the scope of this paper. In this specific ablation, we use `tanh` activations instead of `relu`, solely as a tool to make the intended comparison possible.

Figure 8 shows the results of our experiment. The performance of SAC without target networks supports the common intuition that target networks indeed slow down learning to a small extent. We find that the combination of BatchNorm and Target Networks performs inconsistently, failing to learn anything in half of the environments. Lastly, the configuration of BatchNorm without target networks—and the closest to CrossQ—achieves the best aggregate performance, with the boost being significantly bigger than that from removing target networks alone. In summary, even though removing target networks may slightly improve performance in some environments, it is the combination of removing target networks and adding BatchNorm that accelerates learning the most.

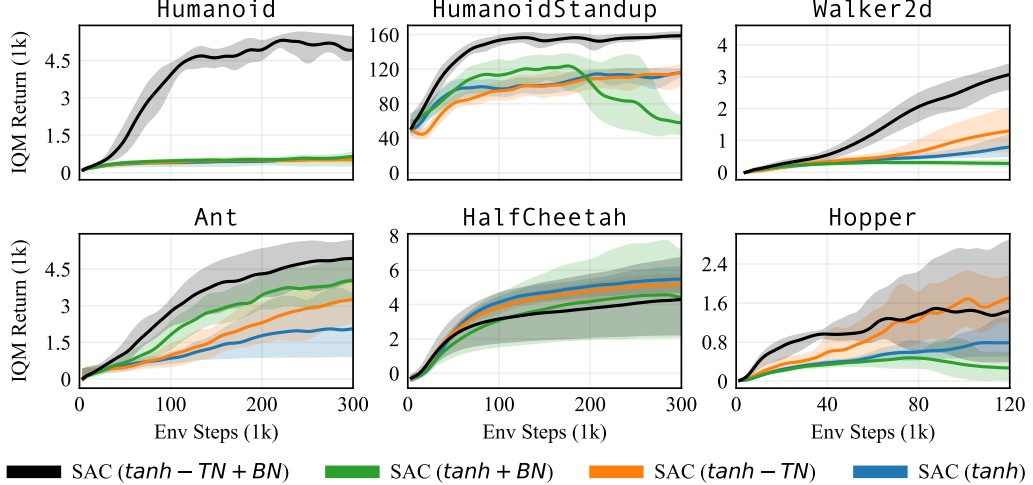

Figure 8: **The effects of target networks and BatchNorm on sample efficiency.** All SAC variants in this experiment use critics with tanh activations, since they allow divergence-free training without target networks, enabling this comparison. This ablation uses the original BatchNorm (BN, Ioffe & Szegedy (2015)). Removing target networks (-TN) provides only small improvements over the SAC baseline with target nets. BatchNorm with target nets (+BN, green) is unstable. Using BatchNorm after removing target nets (-TN+BN)—the configuration most similar to CrossQ—performs best. We plots IQM return and 70% quantile intervals over 10 seeds.

#### 4.4.2 ABLATING THE DIFFERENT DESIGN CHOICES AND HYPERPARAMETERS

In this subsection, we examine the contributions of the different CrossQ design choices to show their importance. Figure 9 shows aggregated ablations of these components and various hyperparameters, while Figure 10 ablates the BatchNorm layer itself.

**Hyperparameters.** CrossQ uses the best hyperparameters obtained from a series of grid searches. Of these, only three are different from SAC's default values. First, we find that reducing the $\beta_1$ momentum for the Adam optimizer (Kingma & Ba, 2015) from 0.9 to 0.5 as well the *policy delay* of 3 have the smallest impact on the performance. However, since fewer actor gradient steps reduce compute, this setting is favorable. Second, reducing the critic network's width to 256—the same small size as SAC—reduces performance and yet still significantly outperforms SAC. This suggests that practitioners may be able to make use of a larger compute budget, i.e., train efficiently across a range of different network sizes, by scaling up layer widths according to the

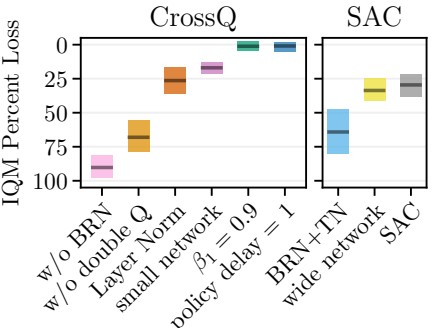

Figure 9: **Ablations on CrossQ and SAC.** Loss in IQM return in percent—relative to CrossQ—at 1M environment interactions. Aggregated over all environments and six seeds each, with 95% bootstrapped confidence intervals (Agarwal et al., 2021). Left shows CrossQ ablations; Right shows effects of adding parts on top of SAC. Figure 13 in Appendix shows individual training curves.

available hardware resources. Third, as expected, removing the BRN layers proves to be detrimental and results in the worst overall performance. A natural question that comes to mind is whether other normalization strategies in the critic, such as Layer Normalization (LayerNorm, Ba et al. (2016)), would also give the same results. However, in our ablation, we find that replacing BatchNorm with LayerNorm degrades CrossQ's performance significantly, roughly to the level of the SAC baseline. Lastly, SAC does not benefit from simply widening critic layers to 2048. And naively adding BRN to SAC while keeping the target networks proves detrimental. This finding is in line with our diagnosis of mismatched statistics being detrimental to the training.

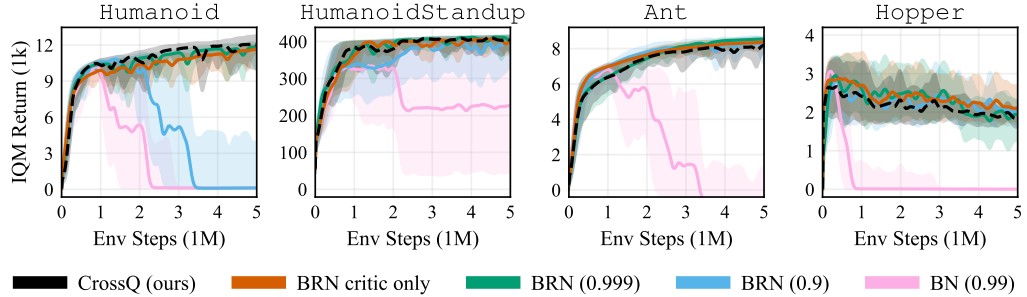

Figure 10: **Comparing BatchNorm hyperparameters.** All variants have comparably strong and stable curves early in the training. Omitting normalization in the actor (BRN critic only) does not significantly affect CrossQ. Using the original Batch Normalization (BN, with moving-average momentum 0.99) is prone to sudden performance collapses during longer training runs. Using BRN permits stabler training, which improves with higher momentums; CrossQ's default 0.99 (black) and higher show no collapses. We plot IQM return and 70% quantile intervals over five seeds.

**Batch Normalization Layers.** In Figure 10, we ablate the BatchNorm versions (BN (Ioffe & Szegedy, 2015) and BRN (Ioffe, 2017)) and their internal moving-average momentums. Compared to CrossQ's optimal combination—BRN with momentum 0.99—all variants have similar sample efficiency in the early stages of training (1M steps). When using BN, we sometimes observe sudden performance collapses later in training; we attribute these to BN's unique approach of using noisy *minibatch estimates* of normalization moments. BRN's improved approach of using the less noisy *moving-averages* makes these collapses less likely; further noise-reduction via higher momentums eliminates these collapses entirely. Additionally, we find that using BatchNorm only in the critic (instead of both the actor and the critic) is sufficient to drive the strong performance of CrossQ; however, including it in both networks performs slightly better.

## 5 CONCLUSION & FUTURE WORK

We introduced CrossQ, a new off-policy RL algorithm that matches or exceeds the performance of REDQ and DroQ—the current state-of-the-art on continuous control environments with state observations—in terms of sample efficiency while being multiple times more computationally efficient. To the best of our knowledge, CrossQ is the first method to successfully use BatchNorm to greatly accelerate off-policy actor-critic RL. Through benchmarks and ablations, we confirmed that target networks do indeed slow down training and showed a way to remove them without sacrificing training stability. We also showed that BatchNorm has the same accelerating effect on training in Deep RL as it does in supervised deep learning. The combined effect of removing target networks and adding BatchNorm is what makes CrossQ so efficient. We investigated the relationship between the Q estimation bias and the learning performance of CrossQ, but did not identify a straightforward dependence. This indicates that the relationship between the Q estimation bias and the agent performance is more complex than previously thought.

In future work, it would be interesting to analyze the Q estimation bias more extensively, similar to Li et al. (2022). Furthermore, a deeper theoretical analysis of the used BatchNorm approach in the context of RL would be valuable, akin to the works in supervised learning, e.g., Summers & Dinneen (2020). Although the wider critic networks do provide an additional performance boost, they increase the computation cost, which could potentially be reduced. Finally, while our work focuses on the standard continuous control benchmarking environments, a logical extension would be applying CrossQ to a real robot system and using visual observations in addition to the robot state. Techniques from image-based RL, such as state augmentation (Laskin et al., 2020; Yarats et al., 2021) and auxiliary losses (Schwarzer et al., 2021; He et al., 2022), also aim to learn efficiently from limited data. We believe some of these ideas could potentially be applied to CrossQ.

ACKNOWLEDGMENTS

We acknowledge the grant "Einrichtung eines Labors des Deutschen Forschungszentrum für Künstliche Intelligenz (DFKI) an der Technischen Universität Darmstadt" of the Hessisches Ministerium für Wissenschaft und Kunst. This research was also supported by the Research Clusters "The Adaptive Mind" and "Third Wave of AI", funded by the Excellence Program of the Hessian Ministry of Higher Education, Science, Research and the Arts, Hessian.AI and by the German Research Foundation (DFG): 417962828.

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

# A APPENDIX

## A.1 DEEPMIND CONTROL SUITE EXPERIMENTS

Figure 11 presents an additional set of experiments performed on the DeepMind Control Suite (Tassa et al., 2018). The experiments shown here are an extension to the experiments shown in Figure 5 in the main paper and have been moved to the Appendix due to space constraints. For the presented tasks, we lowered the learning rate to $8 \times 10^{-4}$ for all algorithms, and set the CrossQ policy delay to 1. All other hyperparameters remained the same as for the main paper.

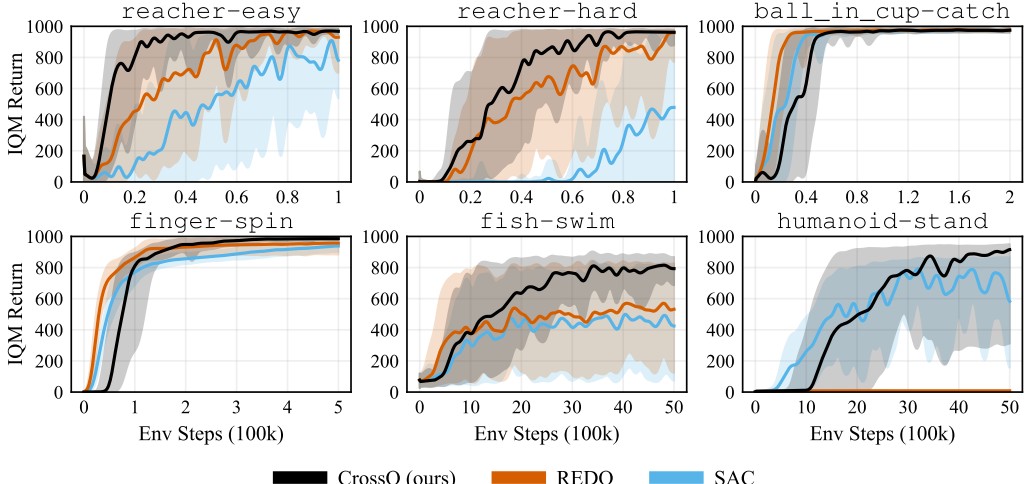

Figure 11: **Sample efficiency of Cross*Q* on DeepMind Control.** The experiments here were each performed on 5 different random seeds. Cross*Q*'s good sample efficiency transfers well to the presented tasks from the DeepMind Control Suite.

## A.2 HYPERPARAMETERS

Experiment hyperparameters, used in the main paper. We adapted most hyperparameters that are commonly used in other works (Haarnoja et al., 2018b; Chen et al., 2021; Hiraoka et al., 2021). The Moving-Average *Momentum* corresponds to 1 minus the Moving-Average *Update Rate* as defined in both BatchNorm papers (Ioffe & Szegedy, 2015; Ioffe, 2017).

Table 1: Learning Hyperparameters

| Parameter | SAC | REDQ | DroQ | Cross$Q$ (ours) |
|---|---|---|---|---|
| Discount Factor ($\gamma$) | | 0.99 | | |
| Learning Rate (Actor & Critic) | | 0.001 | | |
| Replay Buffer Size | | $10^6$ | | |
| Batch Size | | 256 | | |
| Activation Function | | `relu` | | |
| Layer Normalization | No | | Yes | No |
| Dropout Rate | N/A | | 0.01 | N/A |
| BatchNorm / Version | N/A | | | BRN |
| BatchNorm / Moving-Average Momentum | N/A | | | 0.99 |
| BatchNorm / BRN Warm-up Steps | N/A | | | $10^5$ |
| Critic Width | 256 | | | 2048 |
| Target Update Rate ($\tau$) | 0.005 | | | N/A |
| Adam $\beta_1$ | 0.9 | | | 0.5 |
| Update-To-Data ratio (UTD) | 1 | 20 | | 1 |
| Policy Delay | 1 | 20 | | 3 |
| Number of Critics | 2 | 10 | | 2 |

## A.3 WALLCLOCK TIME MEASUREMENT

Wallclock times were measured by timing and averaging over four seeds each and represent *pure training times*, without the overhead of synchronous evaluation and logging, until reaching $5 \times 10^6$ environment steps. The times are recorded on an `Nvidia RTX 3090 Turbo` with an `AMD EPYC 7453` CPU.

Table 2: **Wallclock times.** Evaluated for Cross$Q$ and baselines across environments in hours and recorded on an `RTX 3090`, the details of the measurement procedure are described in Appendix 4.2. Comparing Cross$Q$ with Cross$Q$ (Small) and SAC, it is apparent that using wider critic networks does come with a performance penalty. However, compared to REDQ and DroQ, one clearly sees the substantial improvement in Wallclock time of Cross$Q$ over those baselines.

| | Wallclock Time [hours] | | | | |
|---|---|---|---|---|---|
| | SAC | Cross$Q$ (small) | **Cross$Q$ (ours)** | REDQ | DroQ |
| `HumanoidStandup-v4` | 1.5 | 2.1 | 2.2 | 8.7 | 7.5 |
| `Walker2d-v4` | 0.9 | 0.9 | 1.1 | 4.0 | 4.1 |
| `Ant-v4` | 0.9 | 1.2 | 1.5 | 4.7 | 4.7 |
| `HalfCheetah-v4` | 0.8 | 1.2 | 1.5 | 4.1 | 4.4 |
| `Hopper-v4` | 1.0 | 1.1 | 1.3 | 4.1 | 4.2 |

## A.4 Evolving Action Distributions

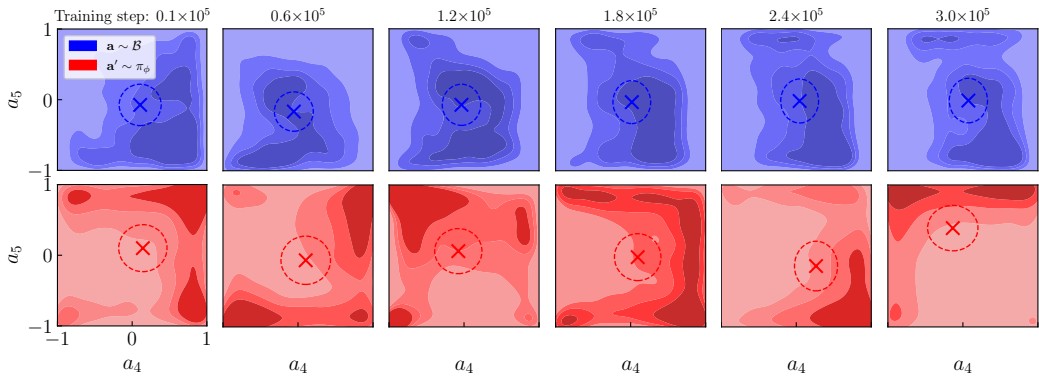

Figure 12: **Replay and policy action distributions are different, and evolve during training.** We train an agent for $300{,}000$ steps on Walker2d. We take snapshots of the replay buffer $\mathcal{B}$ and policy $\pi_\phi$ every $60{,}000$ steps. For each snapshot (one column), we sample a large batch of $10{,}000$ transitions $(s, a, s', a' = \pi_\phi(s'))$ and use this to compute a visually interpretable 2D kernel density estimate of the distributions of $a$ (blue) and $a'$ (red), as seen through the action-space dimensions $4$ and $5$. The cross denotes the mean, and the dashed ellipse is one standard deviation wide for each of the two dimensions. We observe that the distributions as well as the means and standard deviations of the off-policy and on-policy actions are visibly and persistently different throughout the training run, and keep drifting as the training progresses. This discrepancy implies that BatchNorm must be used with care in off-policy TD learning.

### A.4.1 ABLATING THE DIFFERENT DESIGN CHOICES AND HYPERPARAMETERS

Figure 13 depicts in detail the Cross$Q$ and SAC ablations, previously shown in aggregate form by Figure 9.

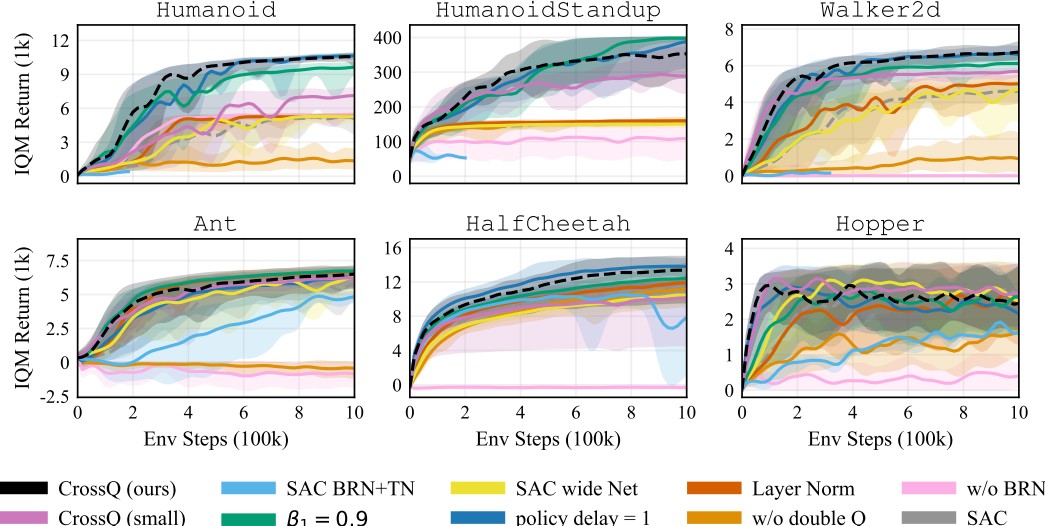

Figure 13: **Cross$Q$ ablation study.** We ablate across different hyperparameter settings and architectural configurations. Using the same network width as SAC, Cross$Q$ (small) shows weaker performance, yet is still competitive with Cross$Q$ in four out of six environments. At the same time, SAC with a wider critic does not work better. Using the default Adam momentum $\beta_1 = 0.9$ instead of $0.5$ degrades performance in some environments. Using a policy delay of 1 instead of 3 has a very small effect, except on Ant. Using LayerNorm instead of BatchNorm results in slower learning; it also trains stably without target networks. Removing BatchNorm results in failure of training due to divergence. Adding BatchNorm to SAC and reusing the live critic's normalization moments in the target network fails to train. Training without double Q networks (single critic) harms performance.

## A.5   REDQ AND DROQ ABLATIONS

Figures 14 and 15 show REDQ and DroQ ablations on 5 seeds each. They show both baselines with the Cross$Q$ hyperparameters: wider critic networks as well as $\beta_1 = 0.5$. Neither baseline benefits from the added changes. In most cases, the performance is unchanged, while in some cases, it deteriorates. The dashed black line shows Cross$Q$ as a reference.

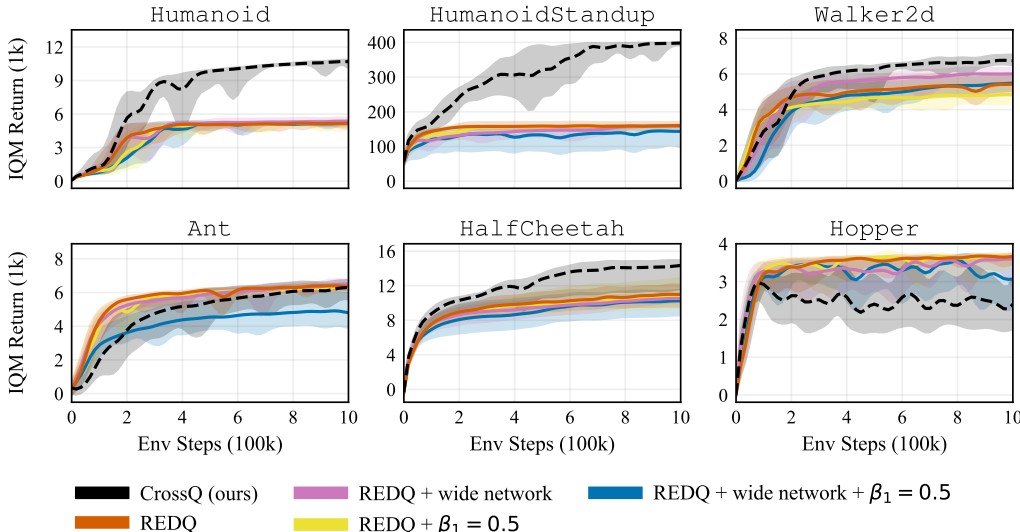

Figure 14: **REDQ ablation.** Showing performance for different combinations of the Cross$Q$ hyper-parameters. The changes in hyperparameters do not help REDQ to get better performance. In fact, in some cases, they even hurt the performance.

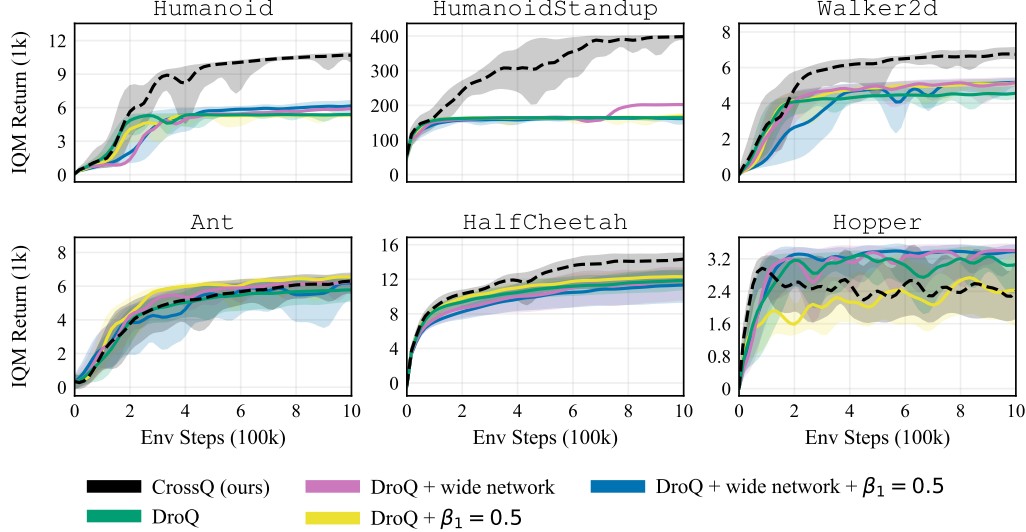

Figure 15: **DroQ ablation.** The changes in hyperparameters do not help DroQ to get better perfor-mance overall. In `Hopper` and `Ant`, performance rises to the Cross$Q$ performance, however, on the `Humanoid`, it hurts performance.

A.6    Effect of Activations and Normalizers on Learning Stability

Figure 8 depicts a small exploratory experiment in which we remove target networks from SAC, and train it with different activation functions and feature normalizers. We do this only to explore whether the boundedness of activations has an influence on training stability. We learn from this experiment that SAC with tanh activations trains without divergence, allowing us to conduct the study in Section 4.4.1. We also observe that at least two feature normalization schemes (on top of the unbounded relu activations) permit divergence-free optimization.

For vectors $\boldsymbol{x}$, relu_over_max($\boldsymbol{x}$) denotes a simple normalization scheme using an underlying unbounded activation: relu($\boldsymbol{x}$)/max($\boldsymbol{x}$), with the maximum computed over the entire feature vector. layernormed_relu simply denotes LayerNorm applied *after* the relu activations. Both of these schemes prevent divergence. Using LayerNorm *before* the relu activations also prevent divergence, and is already explored in the ablations in Figure 13. None of these normalizers perform as strongly as BatchNorm.

A thorough theoretical or experimental study of how activations and normalizers affect the stability of Deep RL is beyond the scope of this paper. We hope, however, that our observations help inform future research directions for those interested in this topic.

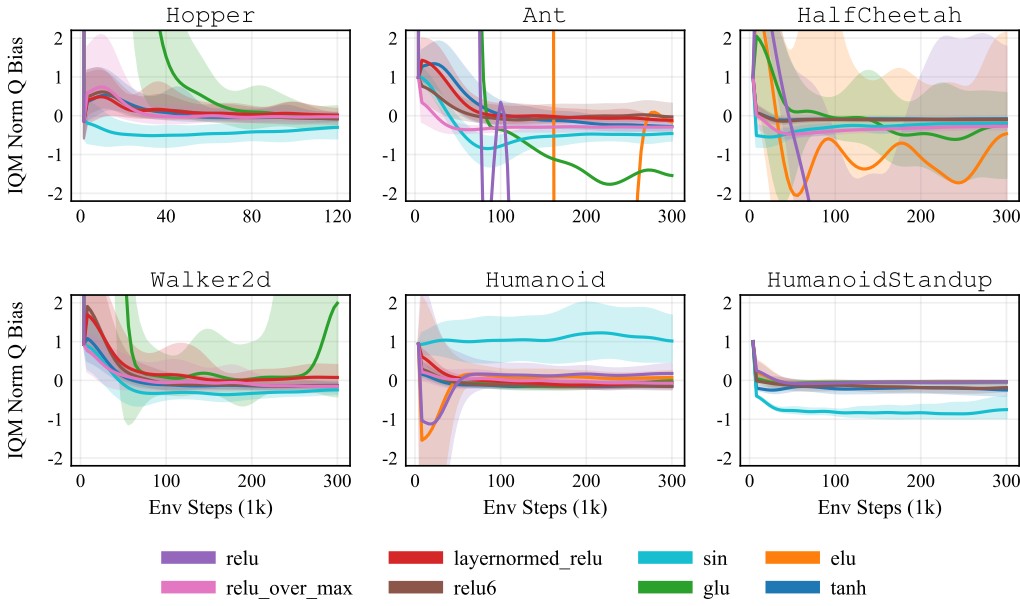

Figure 16: **(In)stability of SAC without target networks.** Observed through the Q estimation bias. In this small-scale experiment, we run SAC with unbounded (relu, glu, elu) and bounded (tanh, relu6, sin) activation functions, as well as "indirectly" bounded activations through the use of two custom normalizers other than BatchNorm (relu_over_max, layernormed_relu). SAC variants with unbounded activations appear highly unstable in most environments, whereas the variants with bounded activations (as well as the normalizers) do not diverge, maintaining relatively low bias.

## A.7 NORMALIZED $Q$ BIAS PLOTS

Figure 17 shows the results of the Q function bias analysis for all environments.

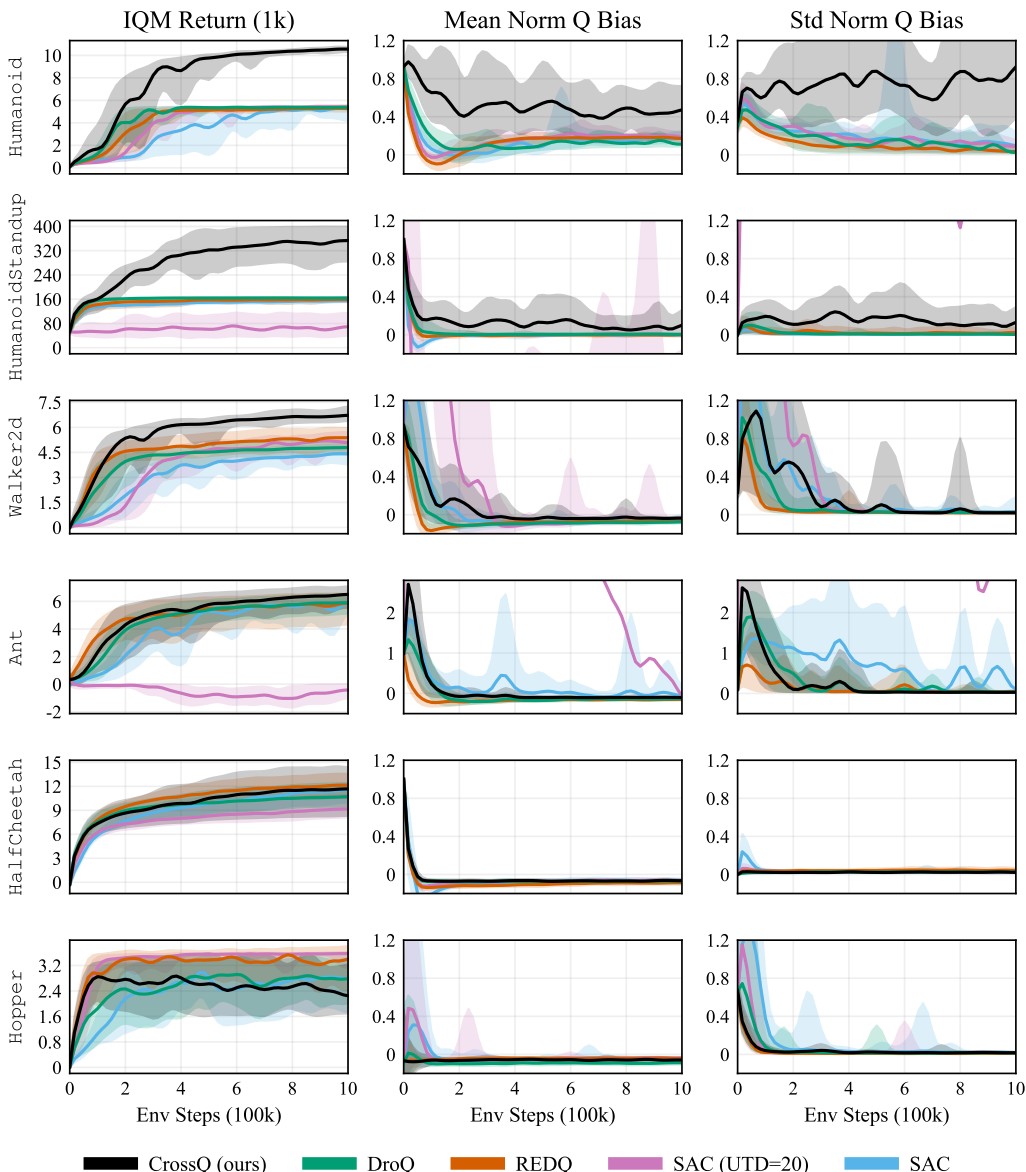

Figure 17: **Q estimation bias.** Mean and standard deviation of the normalized Q function bias, computed as described by Chen et al. (2021). As in the main paper, we do not find a straightforward connection between normalized Q function bias and learning performance. Cross$Q$ generally shows the same or larger Q estimation bias compared to REDQ but matches or outperforms REDQ in learning speed, especially on the challenging `Humanoid` tasks.

