# OpenReview forum: "CrossQ: Batch Normalization in Deep Reinforcement Learning for Greater Sample Efficiency and Simplicity"
_ICLR.cc/2024/Conference — ICLR 2024 spotlight_

### Official Review · Reviewer_7EH3 · 2023-10-23

**Soundness:** 2 fair
**Presentation:** 3 good
**Contribution:** 2 fair
**Rating:** 6
**Confidence:** 4

**Summary:**

The authors present CrossQ, a variation of SAC that removes target networks and leverages Batch Normalization to improve learning efficiency while avoiding high update-to-data ratios. CrossQ presents a straightforward modification of SAC and is experimentally validated on a selection of tasks from the OpenAI Gym.

**Strengths:**

-	The changes over SAC are straight-forward and easy to implement, simplicity being a great asset in RL
-	The approach removes target network dynamics as a step towards decluttering RL
-	Improved learning efficiency over the baselines on the selected tasks
-	Several interesting ablation studies

**Weaknesses:**

- The evaluation looks promising but some experiments are missing to yield a well-rounded study. For example, CrossQ updates the Adam momentum and critic width compared to REDQ and DroQ. How do REDQ and DroQ compare to CrossQ when we set their (1) Adam momentum to 0.5, (2) critic width to 2048, (3) momentum to 0.5 and width to 2048? To reduce compute requirements, this could be studied for e.g., 5 seeds on Humanoid + HumanoidStandup + 1 other task
- The concept should be sufficiently general to combine with other baseline algorithms than SAC, so evaluation across multiple algorithms would further broaden applicability [i.e.are the results specific to SAC variations?]
- Generally, a more diverse set of environments would further strengthen the evaluation – this could include other domains (DeepMind Control Suite / MetaWorld / …) or visual control tasks [i.e.are the results specific to OpenAI Gym domains?]

**Questions:**

- How does CrossQ compare to REDQ and DroQ under the experiments mentioned above?
- Have you tried removing the double Q functions on top of removing the target networks?
- Have you tried keeping the target network and normalize with batch norm parameters of the regular network?
- Do you have an intuition for why CrossQ (black line) performs worse on HalfCheetah in Figure 7?

---

> ### Author Response · Authors · 2023-11-21
> **Answer to Reviewer 7EH3**
>
> We thank the reviewer for suggesting valuable additional ablations and experiments. We added the new results to the updated paper, and we provide our detailed answers below.
>
> ## 1. Additional REDQ and DroQ Ablations
> > The evaluation looks promising but some experiments are missing to yield a well-rounded study. For example, CrossQ updates the Adam momentum and critic width compared to REDQ and DroQ. How do REDQ and DroQ compare to CrossQ when we set their **(1) Adam momentum to 0.5, (2) critic width to 2048, (3) momentum to 0.5 and width to 2048**? To reduce compute requirements, this could be studied for e.g., 5 seeds on Humanoid + HumanoidStandup + 1 other task
>
> These are all good suggestions. We performed each of these experiments on 5 seeds for all environments, and added them to a dedicated section in the Appendix (Figures 14 & 15). We find that **these changes do not improve the baselines’ performance**, and therefore, we retain the better performing versions of the baselines in the main body of the paper.
>
> ## 2. Generality of the Proposed Method
> > The concept should be sufficiently general to combine with **other baseline algorithms than SAC**, so evaluation across multiple algorithms would further broaden applicability [i.e.are the results specific to SAC variations?]
>
> This is an interesting observation. We based CrossQ on SAC because our baselines REDQ and DroQ are also based on SAC. However, our concepts are directly applicable to other off-policy TD algorithms. For demonstration, **we added a TD3 [1] baseline**, as well as a version of CrossQ based on TD3 to Figure 5. This combination shows the same effects in terms of increased sample efficiency, confirming that our proposed design choices regarding batch normalization and target networks can be applied more broadly and are not specific to SAC and the max-entropy RL setting
>
> [1] Scott Fujimoto, Herke Hoof, and David Meger. Addressing function approximation error in actor-critic methods. In International conference on machine learning, 2018.
>
> ## 3. Additional Environments
> > Generally, a more **diverse set of environments** would further strengthen the evaluation – this could include other domains (DeepMind Control Suite / MetaWorld / …) or visual control tasks [i.e.are the results specific to OpenAI Gym domains?]
>
> To align our statements with the experiments, we have narrowed the claims within the paper by specifying more exactly that our results are restricted to continuous-control tasks with state observations. In our original submission, we included the same benchmark suite as the REDQ and DroQ papers, and even reported experiments on 6 environments rather than 4 as in the previous papers. In our updated submission, we double our number by including **6 additional environments on the DeepMind Control Suite**, revealing the same trend in the improved sample efficiency provided by CrossQ. However, note that we had a limited time during the rebuttal to tune CrossQ as well as the baselines on those environments, and better results may be achievable till the camera-ready submission deadline.
>
> We note that a thorough evaluation of CrossQ on vision-based tasks is in general beyond the scope of the present paper and will constitute a separate publication. In the current paper, we focus on establishing CrossQ as a state-of-the-art algorithm for systems with state-based observations, closely following the setting of REDQ and DroQ. **We foresee an extension of CrossQ to vision-based tasks as the next step for future work.**
>
> ## 4. Additional CrossQ Ablations
> > Have you tried **removing the double Q functions** on top of removing the target networks?
>
> We have added this experiment to the ablation study in Fig. 9, however, we find that removing the double Q function in fact hurts the performance.
>
> > Have you tried keeping the target network and normalize with batch norm parameters of the regular network?
>
> We added this experiment to the ablation study in Fig. 9 and found that this configuration does not work. It is very similar to the other, naive, configuration (Fig. 8) of using a target network + BatchNorm. We suspect that the same hypothesis could be used here, which is that the normalization parameters used from the live network do not take into account the on policy actions.
>
> ## 5. CrossQ Bounded Activations Ablation
> > Do you have an intuition for why CrossQ (black line) performs worse on HalfCheetah in Figure 7?
>
> That is because the black-line curves in Figure 8 (previously Figure 7) are not CrossQ, they are only the SAC (tanh) configuration most resembling CrossQ. Figure 8 presents a controlled experiment using tanh activation functions everywhere, because tanh activations let us analyze the effect of the other components on SAC without the presence of target networks.
>
> ---
> We hope this addresses the reviewers concerns. If the reviewer has no additional concerns, we would kindly ask them to increase the score.

---

> > ### Comment · Reviewer_7EH3 · 2023-11-22
> > **Response to rebuttal**
> >
> > Thank you very much for your thorough responses. I believe the updated evaluations have strengthened the paper. I agree that the authors went beyond evaluations presented in prior work, while I'd argue that the domain variety in REDQ and DroQ wasn't particularly broad. The additional experiments are very helpful, I'll raise my score!

---

> > > ### Author Response · Authors · 2023-11-22
> > > **Thank You**
> > >
> > > We thank the reviewer for the kind remarks regarding the rebuttal and the paper, their helpful review and for updating their score.

---

### Official Review · Reviewer_e62k · 2023-10-30

**Soundness:** 3 good
**Presentation:** 4 excellent
**Contribution:** 2 fair
**Rating:** 8
**Confidence:** 5

**Summary:**

The authors address the efficiency of Soft Actor-Critic (SAC) and related deep RL methods for continuous control. Recent methods such as REDQ and DroQ have large update-to-data (UTD) ratios when training the critic (as high as 20). The authors introduce a new method called CrossQ that

1. reduces the critic's UTD to 1,
1. uses batch normalization,
1. removes target networks, and
1. increases hidden-layer widths.

It is shown that these changes are still able to achieve high sample efficiency in MuJoCo environments while being significantly cheaper to run. It is also hypothesized that two of the architectural changes (batch norm and no target networks) are what enable stable training with such a low UTD ratio.

**Score raised from 5 to 8 during rebuttal.**

**Strengths:**

- Good empirical results in MuJoCo. The tested baselines, REDQ and DroQ, are recent and strong baselines yet CrossQ still performs equally or better in the tested environments.
- Removes the target networks of SAC. This is a pain point in terms of both learning speed and implementation, and this is the first time (that I know of) that it has been removed without hurting performance. The authors provide an interesting and sensible hypothesis about the interaction between batch normalization and target networks in Section 3.2 to support their empirical observations. It would be exciting if this generalizes to other algorithms as well.
- The total computation and wall-clock execution time of the proposed method is significantly less than baseline methods, yet sample efficiency is not sacrificed. Along with its simplicity, the new method is very appealing.
- The paper is well written and the discussion of recent related papers is excellent.

**Weaknesses:**

- The paper claims it “show[s] the first successful application of BatchNorm in off-policy Deep RL.” However, it appears that DDPG [1] also used batch normalization 7 years ago in the exact same setting: off-policy deep RL for continuous control in MuJoCo. This means the titular contribution of the paper is not original.
- There is a lot of hyperparameter tuning that might be unfairly benefiting CrossQ. For instance, the network width for CrossQ is increased from 256 to 2048 units, an 8x increase. Adam’s momentum parameter was also reduced from 0.9 to 0.5. As far as I can tell, these hyperparameters were not tested for the baselines, so CrossQ has been over-optimized for MuJoCo. A fairer comparison would be testing the methods with the same network sizes and then selecting the best Adam hyperparameters for each of them.
- I feel that the claim of “state-of-the-art” performance is a bit strong given that only 6 MuJoCo environments were tested, and CrossQ is a clear improvement only in 2 of them. The method has not been evaluated in other continuous-control problems nor against other potential baselines, so I would not necessarily agree that it is state-of-the-art. Nevertheless, the empirical results are still great.

**Minor Edits**
- Inconsistent use of hyphens should be fixed. For example, “Q-function” and “Q function” both appear in the paper.
- Possible typo in Figure 2’s caption: “a **single double** Q function network”?
- As Equation 1 is currently written, it is not correct to refer to the Polyak averaging parameter $\tau$ as “momentum” since $\tau=1$ corresponds to a moving average with no inertia.
- References are not consistently formatted, e.g., some conference names are capitalized but others are not.

**Questions:**

1. At the end of the introduction, you say that your success with batch norm “contradicts” another paper [2] that did not find batch norm to work well. Why do you think you were able to achieve better results? Is it because you removed the target network, or is there another reason?
1. Figure 2’s caption says, “this removes the computational need for two individual forward passes through the same network” in reference to batching the observations and next observations together. But this doesn’t actually reduce computational cost, does it? The same number of forward passes are being done either way.
1. Could you explain the evaluation procedure in the Q-estimation bias experiment (Figure 6)?
1. Could you also explain what is meant by a policy delay of 3? How is this different from the UTD ratio?

**References**

[1] Timothy P Lillicrap, Jonathan J Hunt, Alexander Pritzel, Nicolas Heess, Tom Erez, Yuval Tassa, David Silver, and Daan Wierstra. Continuous control with deep reinforcement learning. 2016.

[2] Takuya Hiraoka, Takahisa Imagawa, Taisei Hashimoto, Takashi Onishi, and Yoshimasa Tsuruoka. Dropout Q-functions for doubly efficient reinforcement learning. 2021.

---

> ### Author Response · Authors · 2023-11-21
> **Answer to Reviewer e62k 1/3**
>
> We thank the reviewer for a lot of thorough questions and suggestions for improvement. We incorporated all the feedback into the updated paper and provided our detailed answers below.
>
> ## 1. Generality of the Proposed Method
> > It would be exciting if this generalizes to other algorithms as well.
>
> This is an interesting observation. We based CrossQ on SAC because the baselines REDQ and DroQ are also based on SAC. However, our **concepts are directly applicable to other off-policy TD algorithms**. For demonstration, we added a TD3 baseline, as well as a **version of CrossQ based on TD3** to Figure 5. This combination shows the **same effects in terms of increased sample efficiency**, confirming that our proposed design choices regarding batch normalization and target networks can be applied more broadly and are not specific to SAC and its max-entropy RL setting.
>
> ## 2. How BatchNorm Was Used in DDPG
> > The paper claims it “show[s] the first successful application of BatchNorm in off-policy Deep RL.” However, it appears that DDPG [1] also used batch normalization 7 years ago in the exact same setting: off-policy deep RL for continuous control in MuJoCo. This means the titular contribution of the paper is not original.
>
> It is true that the DDPG paper reported experiments with batch normalization. However, we would like to emphasize that ours is arguably the first “successful” application of BatchNorm in Deep RL. There are two crucial points:
> 1. In the DDPG paper, **BatchNorm does not significantly improve** the performance, and in the **absence of the target networks even actively harms the performance** (see Fig. 2 in https://arxiv.org/abs/1509.02971). Our paper is the first to use BatchNorm to effectively accelerate Deep RL training by an order of magnitude, similar to supervised learning.
> 2. The DDPG paper’s **use of BN is limited**: it relies on a unique architectural choice where **actions are fused late** into the critic, and **applies BN only to early layers** which appear before this fusion.
>     > “In the low-dimensional case, we [i.e., the DDPG authors] **used batch normalization on the state input** and [...] all layers of the Q network **prior to the action input**.” (Lillicrap et. al 2016).
>
> This is a smart way to **avoid dealing with the on- and off-policy action distribution mismatch in BN** — one simply does not use BN with action-informed features — but is unfortunately only applicable to this specific late-fusion architecture, which is not commonly used.
>
> Moreover, we found several online discussions that claim that BatchNorm actively harms training [1][2], and we could not find any open source implementation of DDPG using BatchNorm. Therefore, we argue that our CrossQ algorithm is effectively the first method that provides a working open source implementation of batch normalization in Deep RL that results in improved learning performance.
>
> **We added a clarification about the difference of our approach compared to DDPG in Sec. 3.2 “Design Choice 2: Using Batch Normalization”.**
>
> [1] https://github.com/keras-team/keras-io/issues/198
>
> [2] https://www.alexirpan.com/2017/04/26/perils-batch-norm.html
>
> ## 3. Additional REDQ and DroQ Ablations
> > There is a lot of hyperparameter tuning that might be unfairly benefiting CrossQ. For instance, the network width for CrossQ is increased from 256 to 2048 units, an 8x increase. Adam’s momentum parameter was also reduced from 0.9 to 0.5. As far as I can tell, these hyperparameters were not tested for the baselines, so CrossQ has been over-optimized for MuJoCo. A fairer comparison would be testing the methods with the same network sizes and then selecting the best Adam hyperparameters for each of them.
>
> We would like to note that **CrossQ performs well even with a smaller critic network and Adam’s momentum 0.9**, as shown in Fig. 9 by the “CrossQ (small)” line. Nevertheless, **we added the requested ablations**, running DroQ and REDQ with increased critic networks and smaller Adam’s momentum in Fig. 14 and 15. As can be seen, **DroQ and REDQ perform the same or worse with these hyperparameters**. Therefore, this ablation confirms that these parameters alone are not sufficient to improve upon the baselines, and the other design choices of CrossQ such as BatchNorm and removal of the target network are also important.

---

> ### Author Response · Authors · 2023-11-21
> **Answer to Reviewer e62k 2/3**
>
> ## 4. Additional Environments, Baselines, and Ablations
> > I feel that the claim of “state-of-the-art” performance is a bit strong given that only 6 MuJoCo environments were tested, and CrossQ is a clear improvement only in 2 of them. The method has not been evaluated in other continuous-control problems nor against other potential baselines, so I would not necessarily agree that it is state-of-the-art. Nevertheless, the empirical results are still great.
>
> We have **toned down the claims** within the paper by specifying more exactly that our claims are **restricted to continuous-control tasks with fully observed state**. In our original submission, we followed the prior work, REDQ and DroQ, and used the same benchmark suite, but we even extended the experiments to 6 environments rather than 4 as in the previous papers. In our updated submission, **we included experiments on DeepMind Control Suite**, which show the same trend in the improved sample efficiency provided by CrossQ. However, note that we had a limited time during the rebuttal to tune CrossQ on those environments, and better results may be achievable till the camera-ready submission deadline.
> We added:
> 1. evaluations on **6 additional Deep Mind Control environments**,
> 2. a **TD3 baseline** (Fig. 5),
> 3. additional **ablations** (Fig. 9, 14, 15),
> 4. a version of **CrossQ based on TD3** (Fig. 5).
>
> We hope this demonstrates the more general applicability of our proposed method.
>
> ## 5. Why BatchNorm Works in CrossQ But Not in DroQ
> > At the end of the introduction, you say that your success with batch norm “contradicts” another paper [2] that did not find batch norm to work well. **Why do you think you were able to achieve better results?** Is it because you removed the target network, or is there another reason
>
> Indeed, DroQ reports experiments with BatchNorm, but they find that it does not improve the performance. Unfortunately, the DroQ code release does not include BatchNorm, therefore, we cannot check the exact implementation. However, **we hypothesize that the DroQ authors used BatchNorm in combination with target networks and did not account for the different batch statistics** (Figure 4), essentially creating the setting which fails to train, as demonstrated in our ablations (Figure 9 (light blue)).
>
> We were able to use BatchNorm successfully because **we took care to use identical normalization moments for both (s, a) and (s’, a’) batches**. Normalizing with identical moments is trivially done using a single combined-batch forward pass; this necessitates predicting both Q(s, a) and Q(s’, a’) through the same network, which by design cuts out the target network.
>
> Reviewer 7EH3 suggested another way to normalize with identical moments, which does not entail target network removal: reusing the (s, a) BatchNorm moments from the “live” network within the target network. We added an experiment testing precisely this configuration in Fig 9, which performs very poorly.
>
> We therefore attribute the superior performance of CrossQ to the synergistic combination of identical normalization statistics and target network removal.
>
> ## 6. Joint Forward Pass
> > Figure 2’s caption says, “this removes the computational need for two individual forward passes through the same network” in reference to batching the observations and next observations together. But **this doesn’t actually reduce computational cost, does it?** The same number of forward passes are being done either way.
>
> We have **rewritten** this sentence in the paper **to make it clearer**. We intended it as a small side note and not as a claim that this is where CrossQ gets its efficiency from. It is true that the same number of operations would be performed in a bigger batch, though on a GPU, performing them in a single bigger batch may be slightly more efficient because this can be done within a single CUDA call instead of two consecutive ones. But it is by no means a crucial point, so we clarified it in the updated submission.
>
> ## 7. Q-Estimation Bias Evaluation Procedure
> > Could you **explain the evaluation procedure in the Q-estimation bias experiment** (Figure 6)?
>
> We used the exact same procedure that was proposed in the REDQ paper. They define the **Q-estimation bias** as the averaged difference between the Q function estimate for the (s, a) pairs sampled at each timestep of an evaluation episode, and their corresponding true Monte Carlo return within that episode: $Q(s, a) - \sum_t \gamma^t r_t$. This Q-estimation bias is then additionally normalized by the magnitude of the Monte Carlo returns, to adjust for the effect of increasing returns over the course of training.

---

> ### Author Response · Authors · 2023-11-21
> **Answer to Reviewer e62k 3/3**
>
> ## 8. Policy Delay
> > Could you also explain what is meant by a **policy delay** of 3? **How is this different from the UTD ratio?**
>
> First introduced in the TD3 paper [3], the **policy delay** prescribes the number of **actor update steps** that are **being skipped for every critic update**. With a policy delay of 3 in CrossQ, we still do one critic update per environment sample (i.e. UTD=1) and **only update the policy once every third environment sample**. The UTD ratio, as used in the REDQ and DroQ papers, prescribes the number of critic updates per environment sample.
>
> [3] Scott Fujimoto, Herke Hoof, and David Meger. Addressing function approximation error in actor-critic methods. In International conference on machine learning, 2018.
>
> ## 9. Minor Edits
>
> We thank the reviewer for pointing these out and have corrected them in the updated version of the paper.
>
> ---
> We hope this addresses the reviewers concerns. If the reviewer has no additional concerns, we would kindly ask them to increase the score.

---

> > ### Comment · Reviewer_e62k · 2023-11-23
> >
> > Thanks to the authors for making these updates and answering my questions. I appreciate that the previously strong claims have been toned down in the paper. By acknowledging the past contributions of others (e.g., DDPG + BN) and explicitly describing why previous approaches were unsatisfactory and how your approach improves them, the paper becomes much stronger and more informative about the use of BN in deep RL.
> >
> > The new ablations look good and have helped convinced me that the improved performance is not just the result of other hyperparameters. Overall, the idea of mixing $s$/$s'$ batch statistics in a single forward pass is really interesting and the empirical analysis is extensive. I will therefore raise my score.

---

### Official Review · Reviewer_enbH · 2023-10-31

**Soundness:** 3 good
**Presentation:** 4 excellent
**Contribution:** 3 good
**Rating:** 6
**Confidence:** 3

**Summary:**

The paper proposes very simple adaptations to the well known SAC algorithm to make it more data and especially more compute efficient. While recent works focused on increasing UTD ratios to make more use of the collected data and thus becoming more data efficient at the cost of compute resources, the authors go the opposite way and show that even with UTD=1 competetive data efficiency is possible with only a fraction of the compute. The main ingredients for the new method are the addition of BatchNormalization, the removal of widely believed necessary target networks, different activation functions as well as wider critic networks.

**Strengths:**

The proposed method achieves competitive sample efficiency while needing only a fraction of the compute, as demonstrated in figures 1 & 5. The paper showcases some interesting findings to enable these advantages - target networks are widely believed to be necessary for stable learning, however the authors found a way around them by using BatchNorm in a novel fashion. Since target networks necessarily slow down learning, this is one part that enables the computationally more efficient learning process. The paper is well written and the major points come across clearly.

**Weaknesses:**

The paper makes very broad claims in terms of proposing the new state of the art in both sample and compute efficiency - they might be justified, however only relatively few baselines (DroQ, REDQ, SAC) are used for comparison. For statements like this, a broader comparison might be required.

Further, even though I am no expert on the the achievable performance in the environments used, I know that at least for some of them higher final performances are possible - e.g. on HalfCheetah I have seen policies do much better than 10k (interestingly the D4RL paper even reports 12k for SAC, whereas in Fig 4. it looks like you only get about 7.5k) - even if other algorithms use more data / compute: if the superior efficiency were only attainable for suboptimal performances, that would be quite limiting and needs to be examined. Please clarify

If you could show some evidence on the suspicion that the reason for prior approaches being unable to use BatchNorm effectively is the OOD-ness of the actions from the policy that are not sampled from the batch in the target networks, that would be great as well since it is I think one of the key insights (i.e. visualize the distributions of sampled and policy actions).

**Questions:**

See weaknesses

---

> ### Author Response · Authors · 2023-11-21
> **Answer to Reviewer enbH**
>
> # Answer to enbH
> We thank the reviewer for suggesting insightful additional experiments. We conducted the suggested studies and added the results in the updated paper. Below we discuss our observations. We hope our answer and the updated paper provide strong evidence in support of our original findings about the proposed CrossQ algorithm and increase the reviewer's score.
>
> ## 1. Papers Claims and Broader Analysis
> > The **paper makes very broad claims** in terms of proposing the new state of the art in both sample and compute efficiency - they **might be justified**, however only relatively few baselines (DroQ, REDQ, SAC) are used for comparison. For statements like this, a **broader comparison might be required**.
>
> We have **narrowed the claims** in the paper by specifying more exactly that our results are restricted to continuous-control tasks with state observations. In our original submission, we followed the prior work, REDQ and DroQ, and used the same benchmark suite as them, but we even extended the experiments to 6 environments rather than 4 as in the previous papers.
>
> As requested, we made the comparison broader in two ways:
> - In our updated submission, we have now included **6 additional experiments on the DeepMind Control Suite**, which show the same trend in the improved sample efficiency provided by CrossQ.
> - Regarding the baselines, we further added:
> 	1. a **TD3 baseline**,
> 	2. **additional ablations** on the current baselines, and
> 	3. a version of **CrossQ based on TD3**.
>
> We hope this demonstrates the more general applicability of our proposed method (see Figure 5).
>
> ## 2. Longer Training Runs
> >  if the superior efficiency were only attainable for suboptimal performances, that would be quite limiting and needs to be examined. Please clarify
>
> We initially conducted all our experiments with a limit of 300k environment interactions, because the baseline REDQ and DroQ papers also use this limit. The focus is on sample efficiency on a limited budget of environment interactions, and the baselines are also computationally costly. However, in our updated submission, **we do present longer runs of CrossQ** in the Appendix in Fig. 13, to demonstrate that longer training runs allow for **reaching higher final performance**, e.g., on HalfCheetah, Humanoid or Walker-2D.
>
> ## 3. Action Distributions
> > visualize the distributions of sampled and policy actions
>
> **We have added the requested plot** to the updated paper. Figure 4 now shows the difference in distributions between on-policy actions and those sampled from the replay buffer. In the Appendix, we also extended this visualization in time, showing how both distributions evolve — and remain different! — over the course of a training run.
>
> ---
> We hope this addresses the reviewers concerns. If the reviewer has no additional concerns, we would kindly ask them to increase the score.

---

> > ### Comment · Reviewer_enbH · 2023-11-22
> > **Rebuttal Response**
> >
> > Dear authors,
> >
> > thank you for your response. I appreciate the additional baseline as well as the action distribution visualisations, I believe they add value to your manuscript. I remain hesitant about the final performance issue: E.g. in the original SAC paper, the algorithm achieves 15k return on HalfCheetah after 3 million timesteps - while you provide longer trainings in the revised version, they are only 700k steps long and only achieve roughly 10k return. Since you very broadly claim SotA sample efficiency, you should also show that you can achieve similar performance with fewer interactions - otherwise the claims need to be further narrowed to something like on par sample efficiency & superior compute efficiency under a fixed sample budget.

---

> > > ### Author Response · Authors · 2023-11-22
> > > **Response to Reviewer**
> > >
> > > We thank the reviewer for clarifying their request; we now better understand what the reviewer is asking for.
> > >
> > > ## **SAC Paper Uses Env Specific Hyperparameters**
> > > Please note, that the **SAC paper uses environment-specific hyperparameters** (see Appendix D, Table 2 in the SAC paper https://arxiv.org/abs/2108.13264), **whereas we use identical hyperparameters** for all the gym tasks.
> > >
> > > Unlike the SAC paper, we chose our CrossQ hyperparamters to achieve the **best aggregate performance** across all considered environments; our goal was not to achieve the **maximum possible score on any specific environment** like e.g. HalfCheetah.
> > >
> > > ## **CrossQ Achieves Better Final Score With Far Fewer Interactions in Most Envs**
> > > Despite this, **CrossQ acheives the highest numerical scores on 3 out of 6 gym environments** with identical hyperparameters, including the 2 hardest Humanoid tasks. On **Humanoid, where CrossQ doubles REDQ's final performance** getting to 10k in 700k steps, compared to the SAC paper's 6k in 10M steps (**1.6x score** with less than **1/10 of the environment interactions**).
> > >
> > > **On DeepMind Control Suite, we reach the highest possible scores (1000) on all environments and significantly outperform the baselines** (except Fish where we match the highest score reported in the original DeepMind Control paper https://arxiv.org/pdf/1801.00690.pdf, and Humanoid, which in the short rebuttal duration, we could not get any algorithm to solve).
> > >
> > > In the rebuttal, we have shown that with an increased computation buget, for longer runs, CrossQ benefits and **can obtain even higher scores**. Due to the short rebuttal phase (and the many other additional experiments we now included) we could not present millions of steps on all environments, however, we can include this in the camera ready version.
> > >
> > > ---
> > >
> > > However, if the reviewer maintains their view that HalfCheetah's final performance weakens our claim, we are happy to finetune CrossQ's hyperparameters (and explore environment specific hyperparameters like the referenced SAC paper) to deliver upon their request. Given the time constraints, we would do this finetuning by the camera-ready deadline and update the plots with much longer runs.

---

> > > > ### Comment · Reviewer_enbH · 2023-11-23
> > > > **Response 2**
> > > >
> > > > Dear authors,
> > > >
> > > > thank you for your swift reply and clarification of my question - I was unaware other algorithms achieved the higher performance only through environment specific hyperparameter tuning. Since crossQ performs better than prior works on most of the benchmarks I don't want to be too fixated on the halfcheetah & will raise my score to 6. If you could include long(er) training runs in the camera ready, I would still find it to add value.

---

### Official Review · Reviewer_stzr · 2023-11-01

**Soundness:** 3 good
**Presentation:** 4 excellent
**Contribution:** 3 good
**Rating:** 8
**Confidence:** 5

**Summary:**

The paper presents CrossQ, an algorithm for deep reinforcement learning, aiming to address the crucial issue of sample efficiency. CrossQ achieves state-of-the-art sample efficiency, reduces computational costs significantly compared to existing methods (REDQ and DroQ), and simplifies the implementation process by eliminating the need for advanced bias-reduction schemes and high update-to-data (UTD) ratios. The core innovations of CrossQ include the removal of target networks, strategic use of Batch Normalization, and wider critic networks.

**Strengths:**

1. Well-written paper, easy to follow. The paper is well-structured and clearly articulates the motivation, methodology, and experimental results. The use of figures and code snippets aids in understanding the algorithm's implementation, making it accessible to readers with varying levels of expertise.

2. CrossQ's approach stands out in the field of deep reinforcement learning by challenging the trend of high UTD ratios. The innovative removal of target networks, combined with Batch Normalization and wider critic networks, is a departure from conventional methods, leading to improved sample efficiency and reduced computational complexity.

3. The paper demonstrates a rigorous empirical analysis of CrossQ's performance through ablation studies, clearly showcasing the algorithm's effectiveness. The use of Batch Normalization in a novel manner, coupled with wider networks, highlights a high-quality exploration of the algorithm's design space.

**Weaknesses:**

1. While the empirical results are compelling, the paper lacks theoretical analysis or justification for the success of CrossQ. Providing insights into why the proposed modifications lead to improved performance would enhance the paper's depth and contribute to a more comprehensive understanding of the algorithm.

2. The experiments are only conducted in one continuous control domain (MuJoCo). It is dangerous to draw conclusion of sample efficiency only from one experimental domain.

3. Also, this paper does not include image-based settings. There are tons of efficient RL works recently on image-based settings and some of the techniques (state augmentation [1, 2] and auxiliary losses [3, 4]) can be also applied to state-based settings. I think it is worth discussing the recent progress of efficient RL in the related work session or adding baselines for comparison.

[1] Laskin, M., Lee, K., Stooke, A., Pinto, L., Abbeel, P., & Srinivas, A. (2020). Reinforcement learning with augmented data. Advances in neural information processing systems, 33, 19884-19895.

[2] Yarats, D., Fergus, R., Lazaric, A., & Pinto, L. (2021). Mastering visual continuous control: Improved data-augmented reinforcement learning. arXiv preprint arXiv:2107.09645.

[3] Schwarzer, M., Anand, A., Goel, R., Hjelm, R. D., Courville, A., & Bachman, P. (2020). Data-efficient reinforcement learning with self-predictive representations. arXiv preprint arXiv:2007.05929.

[4] He, T., Zhang, Y., Ren, K., Liu, M., Wang, C., Zhang, W., ... & Li, D. (2022). Reinforcement learning with automated auxiliary loss search. Advances in Neural Information Processing Systems, 35, 1820-1834.

**Questions:**

1. Can you elaborate on the specific scenarios or domains where CrossQ might outperform other algorithms significantly? Are there any limitations or challenges in which CrossQ may not be the most suitable choice?

2. The paper focuses on empirical results; however, are there any insights or intuitions on why the chosen modifications (removal of target networks, Batch Normalization, wider networks) lead to the observed improvements? Providing additional context on the underlying mechanisms would enhance the paper's impact.

3. To me, I think using wider critic networks played a pivot role in the performance boost, especially in humanoid-based environments. Can you draw SAC+wider networks to Figure 7?

I would be happy to raise my scores if my concerns are addressed.

---

> ### Author Response · Authors · 2023-11-21
> **Answer to Reviewer stzr 1/2**
>
> We wish to thank the reviewer for their comments and positive feedback. We believe we have been able to address all their points in the following paragraphs.
>
> ## 1. Insights and Ablations
> > While the empirical results are compelling, the paper lacks theoretical analysis or justification for the success of CrossQ. **Providing insights into why the proposed modifications lead to improved performance would enhance the paper's depth** and contribute to a more comprehensive understanding of the algorithm.
>
> We added **further insights** into why the proposed modifications lead to improved performance in several places in the paper:
> 1. we added Fig. 4 that show the difference in the action distributions from the replay buffer and from the policy, highlighting the **importance of using batch normalization** in the proposed manner;
> 2. in Fig. 9, we added an ablation experiment running SAC with both BatchNorm and Target Networks (as suggested by reviewer 7EH3), showing that our design choice of **removing the target network is important** for the superior performance of the proposed CrossQ algorithm;
> 3. we added Fig. 14 and 15, providing ablations on DroQ and REDQ with elements of CrossQ such as wider critic networks and $\beta\_1 = 0.5$, showing that these changes alone are insufficient to boost their performance, and **it is really the combination of the components in CrossQ** that is responsible for the strong learning performance.
>
> Thus, we tried our best to **provide intuitions and establish general facts** about the proposed design choices. Nevertheless, we admit that a deeper theoretical investigation would be very illuminating, despite being a difficult task. We hope that our paper may stimulate interest in studying the effects of different design choices in Deep RL algorithms.
>
>
> ## 2. Strengths and Weaknesses of CrossQ
> > Can you elaborate on the specific scenarios or **domains where CrossQ might outperform other algorithms** significantly? Are there any **limitations** or **challenges** in which CrossQ may not be the most suitable choice?
>
> This is a very good question!
>
> In general, we believe **CrossQ should outperform algorithms which use low UTD ratios** and use no special techniques to accelerate convergence. E.g. CrossQ outperforms SAC and TD3 significantly, but is comparable in efficiency to REDQ which emphasizes making the most out of the available data.
>
> In addition to the potential limitations we’ve already noted in the conclusion section, we believe CrossQ may inherit any challenges that BatchNorm introduces. For example, models with BatchNorm are known [1] to struggle in **training with very small minibatch sizes**. BatchNorm uses minibatch statistics to normalize data, and while minibatch noise is regarded as a regularizer, it has also been implicated in sometimes harming optimization stability [2]. BatchNorm-equipped training is also sensitive to non-i.i.d batches, and could possibly struggle with non-stationary data sampling schemes such as **Prioritized Experience Replay** [2]. Such limitations in BatchNorm have been addressed with very simple fixes like Batch Renormalization [4], which might also benefit CrossQ in the future.
>
> [1] https://arxiv.org/abs/1708.02002
>
> [2] https://proceedings.mlr.press/v89/lian19a.html
>
> [3] https://github.com/yhenon/pytorch-retinanet/issues/24
>
> [4] https://arxiv.org/abs/1702.03275
>
> ## 3. Additional Environments and Evaluations
> > The experiments are only conducted in one continuous control domain (MuJoCo). It is dangerous to draw conclusion of sample efficiency only from one experimental domain.
>
> We followed the prior work, REDQ and DroQ, and used the same benchmark suite. In fact, even in our original submission we extended the experiments from 4 to 6 environments, adding the hard HumanoidStandup task. Following the reviewer's suggestions, in our updated submission **we have doubled the number of environments** by adding **6 DeepMind Control Suite tasks**. We observe the same trend of improved sample efficiency provided by CrossQ (see Fig. 10). We hope that this addresses the reviewer's concern about the applicability of CrossQ to other environments.

---

> ### Author Response · Authors · 2023-11-21
> **Answer to Reviewer stzr 2/2**
>
> ## 4. CrossQ on Vision-based RL Tasks
> > Also, **this paper does not include image-based settings**. There are tons of efficient RL works recently on image-based settings and **some of the techniques** (state augmentation [1, 2] and auxiliary losses [3, 4]) **can be also applied to state-based settings**. I think it is worth **discussing the recent progress** of efficient RL in the related work session or adding baselines for comparison.
>
> We agree that a thorough evaluation of CrossQ on vision-based tasks would be very interesting; however, we consider this extension as being **beyond the scope of the present paper**. Efficient image-based RL requires multiple ingredients, as noted by the reviewer, and a careful study of CrossQ in tandem with these would constitute a separate publication.
>
> In the current paper, we focus on establishing CrossQ as a state-of-the-art algorithm for continuous control from state observations, closely following the setting of REDQ and DroQ. We do agree that much like our use of BatchNorm, the techniques mentioned by the reviewer are also strong tools for boosting RL efficiency; **we have added the suggested references** and highlighted the extension of CrossQ to **vision-based tasks as the next step for future work**.
>
>
> ## 5. Are Wider Critic Networks Crucial for CrossQ Performance?
> > To me, I think **using wider critic networks played a pivot role** in the performance boost, especially in humanoid-based environments. Can you **draw SAC+wider networks** to Figure 7?
>
> We note that even without the wider critic network, CrossQ delivers a strong performance, as shown in Figure 9 in the updated paper by the magenta curve corresponding to “CrossQ (small)”. We have added the requested SAC+wider network “SAC wide Net” ablation study to Figure 9 (yellow curve). There, we observe that **SAC does not get a significant improvement** simply from adding wider networks, indicating that it is **the combination of CrossQ’s design choices that is important**, and not only the wider critic networks alone.
>
> ---
> We hope this addresses the reviewers concerns. If the reviewer has no additional concerns, we would kindly ask them to increase the score.

---

> > ### Comment · Reviewer_stzr · 2023-11-21
> > **Thank you**
> >
> > Having thoroughly reviewed all comments and responses provided in the reviews, I want to thank the authors for promptly addressing my concerns with additional details. This paper is truly commendable, and I have accordingly adjusted my score to 8.

---

> > > ### Author Response · Authors · 2023-11-22
> > > **Thank you**
> > >
> > > We would like to express gratitude to the reviewer for their kind words and for raising their score.

---

### Author Response · Authors · 2023-11-21
**General Response**

We thank all the reviewers for their positive and extensive feedback, and their constructive questions. We are sorry for the rather late rebuttal, we were hard at work preparing lots of additional experiments and clarifications. We believe we have been able to address all issues pointed out and hope that our responses satisfy all of the reviewers’ requests.

We have **updated the paper** to reflect our improvements and to include the reviewers’ feedback, with the requested changes highlighted in red. We **have uploaded our source code** as part of the supplementary material. Further, we have provide a **general response** here as well as **extensive individual answers** to each reviewer below.

## Positive Impressions
1. We are very pleased that the reviewers acknowledge the **novelty and importance** of our submission:
- `stzr`: CrossQ's approach stands out in the field of deep reinforcement learning [and] is a departure from conventional methods.
- `enbH`: the authors found a way around [target networks] by using BatchNorm in a novel fashion.
- `e62k`: [target networks are] a pain point in terms of both learning speed and implementation, and this is the first time (that I know of) that it has been removed without hurting performance.
- `7eh3`: The approach removes target network dynamics as a step towards decluttering RL.

2. The reviewers agree on the paper’s **good empirical results** on the presented MuJoCo environments:
- `stzr`: The paper demonstrates a rigorous empirical analysis
- `enbH`: The proposed method achieves competitive sample efficiency while needing only a fraction of the compute
- `e62k`: computation and wall-clock execution time [...] is significantly less. [...] Along with its simplicity, the new method is very appealing.
- `7eh3`: Improved learning efficiency over the baselines on the selected tasks

3. We are further pleased about the positive comments regarding the **presentation and clarity** of the paper:
- `stzr`: Well-written paper, easy to follow.
- `enbH`: The paper is well written and the major points come across clearly.
- `e62k`: discussion of recent related papers is excellent.

## Summary of Common Questions
We identified three major points reviewers agreed upon, which we want to address here:

### 1. Broader Scope of Experiments
There were multiple requests for further experiments and studies and that claiming state-of-the-art performance might require more evaluations.

1. Some reviewers requested **additional ablation studies**. We have added those to Section 4.4.2 “Ablating the Design Choices and Hyperparameters”. The findings are in line with our previous results and did not yield any surprises, confirming our key observations about the influence of the components of CrossQ.
2. We further **added a TD3 baseline** (see Fig. 5) and a version of CrossQ that uses TD3 as the base algorithm, without any further hyperparameter tuning. We observe similar performance gains that we showed for SAC, demonstrating that our proposed design choices are general and applicable to different base algorithms, and not only SAC with its max-entropy RL setting.
3. We **added six additional environments** (including 4 non-locomotion tasks) on the DeepMind Control Suite in Fig. 10, which confirm the improved sample efficiency of CrossQ across more tasks in a different domain (DM Control). To make our contribution clearer, we toned down the state-of-the-art claims and specified more explicitly that our results are restricted to continuous-control tasks with state-based observations.
4. While we agree that **vision-based RL** would be an interesting application of our work, we believe that the complexity of pixel-based RL pipelines and the thoroughness required to investigate this different problem setting would necessitate a separate publication.

### 2. CrossQ Hyperparameters
Some reviewers asked for additional studies regarding our chosen hyperparameters, as some CrossQ hyperparameters have different values compared to REDQ and DroQ. We reran REDQ and DroQ experiments with CrossQ’s hyperparameters, and added a corresponding section A.7 to the Appendix. We find that the **hyperparameters which benefit CrossQ actually hurt REDQ and DroQ** performance and that the hyperparameters we initially used for the baselines are in fact the best we could find for those tasks.

### 3. More Insights Into the Method
Some reviewers asked for further insights into why the proposed design choices improve the performance. We added further evidence for our hypotheses. This includes
1. visualizations of the different action distributions in Fig. 4,
2. new ablation experiments in Fig. 9,
3. new Figures 14 and 15 with further ablations of the baselines.

Nevertheless, we admit that a deeper theoretical investigation would be very illuminating, and we hope that our paper may stimulate the interest in studying the effects of different design choices in Deep RL algorithms, despite this being a non-trivial task.

---

### Public Comment · ~Jen-Yen_Chang1 · 2024-04-03
**Question about pseudocode**

Thank you for this great work.
I have a small question regarding the pseudocode in Figure 2. In Line 2, it is written
`next_acts, next_logpi = policy.apply(policy_params, obs)`
I am a bit confused and I would like to believe it to be similarly to SAC as the following:
`next_acts, next_logpi = policy.apply(policy_params, next_obs)`

If the authors will enlighten me, I shall be very grateful!

---

> ### Public Comment · ~Aditya_Bhatt1 · 2024-05-01
> **Good catch**
>
> That should indeed say `next_obs`. Thanks for pointing this out! We have fixed it in the newest update.

---

### Meta-Review · Area_Chair_B2nP · 2023-12-06

**Metareview:**

The paper proposes several additions to the popular soft actor critic (SAC) algorithm that improve the computational efficiency while maintaining the sample efficiency of other recent works based on large update-to-data ratios. The reviewers found the paper to be novel, empirically convincing, and clear. The reviewers raised several concerns about the empirical evaluation, but these were largely addressed by the authors during the rebuttal.This prompted several reviewers to raise their scores significantly.

**Justification For Why Not Higher Score:**

This paper presents a series of design choices on top of SAC that result in improved computational efficiency. The core contribution lies in the set of hyperparameters / empirical setup. While this is an important contribution, I do not believe the level of novelty of the paper warrants an oral.

**Justification For Why Not Lower Score:**

The paper tackles the well-known D4RL benchmark and makes a series of design choices that demonstrate that higher update-to-data ratios are not necessary for improved sample efficiency. This is an important contribution that may be applicable to the broader set of areas where RL is applied.

---

### Decision · Program_Chairs · 2024-01-16

Accept (spotlight)